



# Quantifying energy barriers associated with density stratification in vertical displacements of water parcels

Efraín Moreles[1], Emmanuel Romero[2], and Benjamín Martínez-López[3]

[1]Instituto de Ciencias del Mar y Limnología, Universidad Nacional Autónoma de México, Mexico City, Mexico
[2]Departamento Académico de Sistemas Computacionales, Universidad Autónoma de Baja California Sur, Carretera al sur km 55, Col. Mezquitito, La Paz, BCS, Mexico
[3]Instituto de Ciencias de la Atmósfera y Cambio Climático, Universidad Nacional Autónoma de México, Mexico City, Mexico

**Correspondence:** Efraín Moreles (moreles@cmarl.unam.mx)

**Abstract.**

Density stratification affects mixing by limiting the interchange of properties along the water column. This stratification establishes barriers to the vertical displacement of water parcels, which interchange their properties between different layers of the ocean. Quantifying the intensity of these barriers, considering the direction (upward and downward) of the vertical

displacement of a water parcel, is the main aim of this study. By employing the buoyancy energy required to displace a water parcel vertically, we propose the buoyancy potential energy (BPE) to analyze energy barriers in detail, describe the stability of a water parcel, and provide an objective integrated measure of stratification intensity in any vertical section of the water column. By illustrating the application of BPE in dynamical and ecological contexts, we demonstrate that it provides a complementary understanding of oceanic stratification. BPE is directly calculated from survey data, eliminating the need to

diagnose turbulent processes, and has a straightforward numerical implementation. The routine application of BPE to study diverse ocean-atmosphere interaction processes (e.g., dynamical, thermodynamical, ecological, and biochemical) in which stratification plays a role can extend and enrich our understanding of them.

## 1 Introduction

The vertical variation in density in aquatic bodies (i.e., oceans, seas, freshwater bodies, and estuaries), resulting from temperature and salinity differences, constitutes their stratification, which has significant implications for the variability of weather and climate processes as it determines different dynamical, ecological, chemical, and biological processes occurring there. Stratification has a significant role on the flux of particulate organic matter from the surface to sediments (e.g., Kirillin et al., 2012; Omand et al., 2020), the vertical content of chlorophyll (e.g., Cullen, 2015; Carvalho et al., 2017; Briseño-Avena et al.,

2020; Cornec et al., 2021; Zampollo et al., 2023), biological productivity (e.g., Franks, 2014; Bouman et al., 2020), the mixed





layer (e.g., Sutherland et al., 2014; Gray et al., 2020; Moreles et al., 2025), the ocean-atmosphere coupling and exchanges between them (e.g., Deser et al., 2010; Groeskamp et al., 2019), barrier layers (e.g., Sprintall and Tomczak, 1992; Cronin and McPhaden, 2002), and tropical cyclone dynamics (e.g., Wang et al., 2011; Balaguru et al., 2012; Vincent et al., 2012; Yan et al., 2017). Quantifying stratification and destratification processes is crucial for a deeper understanding of the complex interactions

in aquatic bodies and their variability at both small and large spatiotemporal scales. Stratification determines mixing, which is commonly associated with the interchange of properties between deep and shallow waters, and vice versa. This vertical interchange of properties is typically linked to the vertical displacement of water parcels, which carry their properties from one isopycnal to another. The vertical movement of a water parcel is, to a certain extent, determined by the density stratification experienced by the water parcel when it displaces upward and downward; the more stratified the water column, the more energy

is required to displace the water parcel vertically, and vice versa.

Stratification in aquatic bodies has been intensively studied. Schmidt (1928) derived an equation for the stability of a lake; Idso (1973) subsequently extended that equation to account for the contribution of each lake layer to stability. Simpson et al. (1978) and Herrmann et al. (2008) proposed energy-based indexes to quantify stratification from the surface to a specific depth. Dynamic equations derived from Simpson et al.'s index are used to analyze the evolution of stratification in terms of

different processes (Simpson and Bowers, 1981; Burchard and Hofmeister, 2008; de Boer et al., 2008). While valuable, the above developments do not focus on quantifying the energy required to vertically displace a water column in terms of density stratification at intermediate depths within the water column. The above is particularly relevant when we consider that many physical, ecological, chemical, and biological processes occur in the subsurface or at a distance from the surface. Furthermore, analyzing stratification requires considering the direction in which it is quantified: the stratification experienced by a deep-

water parcel when it displaces upward differs from that experienced by a shallow-water parcel when it displaces downward. A metric that provides information on the energy barriers associated with density stratification at intermediate depths within the water column would complement the analysis of the processes governing the vertical structure in aquatic bodies.

This study aims to fill this gap by deriving an energy-based metric that quantifies the energy required to displace a water parcel vertically, taking into account the density stratification and the direction of the vertical displacement. The metric we

propose quantifies the buoyancy energy required for a water parcel in equilibrium to be displaced upward and downward; this metric describes the stability of the water parcel by identifying its type of equilibrium state. Also, the metric provides an integrated measure of the stratification intensity in any vertical section of the water column. Finally, we present the application of this metric in both dynamical and ecological contexts, demonstrating its potential for enriching ocean analyses.

## 2 Quantifying energy barriers

Consider a fluid in hydrostatic balance in which a water parcel is adiabatically displaced along the vertical from its equilibrium position $z_{eq}$ to any depth $z$. In such a displacement, the parcel experiences a buoyancy force given by (Vallis, 2006, p. 92)

$$F(z) = g\left[\rho^{\theta}(z) - \rho^{\theta}(z_{eq})\right], \tag{1}$$





where $g$ is the acceleration due to gravity, and is the potential density of the environment referred to the pressure at the level $z$. At its equilibrium position, the water parcel does not experience a buoyancy force, $F(z_{eq}) = 0$. For processes occurring in the first tens of hundreds of meters, the potential density should be referred to 0 dbar. The buoyancy force is conservative (it is a function only of the vertical coordinate) and can be written in terms of the vertical derivative of a potential function (BPE),

$$F(z) = -\frac{d}{dz}\text{BPE}(z). \tag{2}$$

BPE is the potential energy function associated with the buoyancy force and is obtained by vertically integrating the buoyancy force in Eq. (2),

$$\text{BPE}(z) = \text{BPE}(z_{eq}) - \int_{z_{eq}}^{z} g\left[\rho^{\theta}(\gamma) - \rho^{\theta}(z_{eq})\right]d\gamma = \text{BPE}(z_{eq}) + g(z - z_{eq})\rho^{\theta}(z_{eq}) - g\int_{z_{eq}}^{z}\rho^{\theta}(\gamma)d\gamma. \tag{3}$$

In energy analyses, the physically relevant quantity is the potential difference between two depths; thus, we can set $\text{BPE}(z_{eq}) = 0$ without loss of generality. BPE is directly related to the work done by buoyancy WB as proposed by Moreles et al. (2025); indeed, $\text{WB}_{z_1 \to z_2} = \text{BPE}(z_1) - \text{BPE}(z_2)$. Thus, all the properties and attributes of WB described in Moreles et al. (2025) are directly applicable to BPE.

BPE represents the buoyancy potential energy required to displace a water parcel from its equilibrium position $z_{eq}$ to any level $z$; it represents the energy barriers the parcel encounters during its vertical displacement. Depending on the sign of BPE, two physical situations are identified. For $\text{BPE} > 0$, the force and the parcel displacement are in opposite directions, causing the parcel to decelerate when it moves. It is the typical behavior in stably stratified columns (Fig. 1a), where BPE is in the shape of a potential well and the parcel is in stable equilibrium. When the parcel is displaced up or down from its equilibrium position (the valley), the buoyancy force tends to return it to its equilibrium position, initiating an oscillatory motion; energy must be supplied to the parcel to displace it vertically. For $\text{BPE} < 0$, the force and the parcel displacement are in the same direction, causing the parcel to accelerate when it moves. It is the typical behavior in unstably stratified columns (Fig. 1b), characterized by BPE in the shape of a hill and the parcel in unstable equilibrium. When the parcel is vertically displaced from its equilibrium position (the hilltop), the buoyancy force pushes the parcel outward down the slope of the hill, initiating a convective movement that does not require additional energy to maintain it. For a water column that is perfectly homogeneous in density, a water parcel can be vertically displaced without experiencing any buoyancy force; the parcel neither returns to its original equilibrium position nor continues to move away from its new position. BPE is zero along the column, indicating that there are no energy barriers to the vertical displacement of the parcel, and the parcel is in a state of neutral equilibrium.

The equilibrium state and the associated BPE profile for a water parcel change along the vertical, depending on the local density structure. Figure 2 shows the BPE profiles at different equilibrium positions for an observed density profile. For example, consider the equilibrium position of 19.9 m and the section from 7 to 80 m. Taking this equilibrium position as a reference, the environmental density is greater upward and lower downward. A parcel at the equilibrium position perceives the water column as unstably stratified, so the parcel will displace with an accelerated motion that does not require additional energy to maintain. However, around the equilibrium position, from 19 to 25 m, the environmental density varies very little. A parcel at



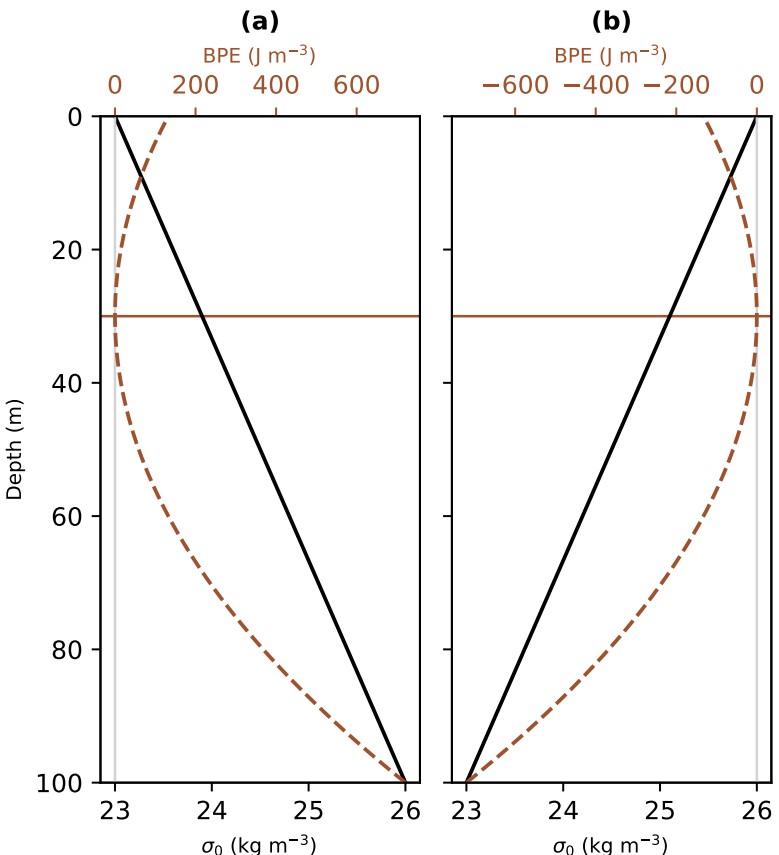

**Figure 1.** Vertical profiles of potential density anomaly ($\sigma_0$) (black solid line) and BPE (brown dashed line) for an equilibrium position at 30 m, considering (a) a stably stratified and (b) an unstably stratified water column. The BPE profile in (a) represents a barrier, whereas in (b) it represents an unobstructed way for the vertical displacement of a water parcel at a depth of 30 m.

the equilibrium position perceives the water column as homogeneous at these depths, so the parcel will displace without requiring additional energy, but without initiating accelerated motion. The behavior of the parcel described above is represented by a negative BPE profile (brown solid line in Fig. 2b) with a local maximum at the equilibrium position, with near-zero values from 19 to 25 m. Finally, for the water column section shallower than 7 m, the environmental density is lower than the density at the equilibrium position. In this upper section, a water parcel with a density equal to that of the environment at a depth of 19.9 m will displace in a stably stratified environment, resulting in BPE no longer decreasing and starting to increase. The other BPE profiles can be described using a similar analysis.

As described above, the vertical structure of density determines the shape of BPE and the consequent stability of a water parcel if it were vertically displaced from its original equilibrium position. BPE thus describes the stability of a water parcel by identifying its type of equilibrium state: stable (BPE is a local or global minimum), unstable (BPE is a local or global





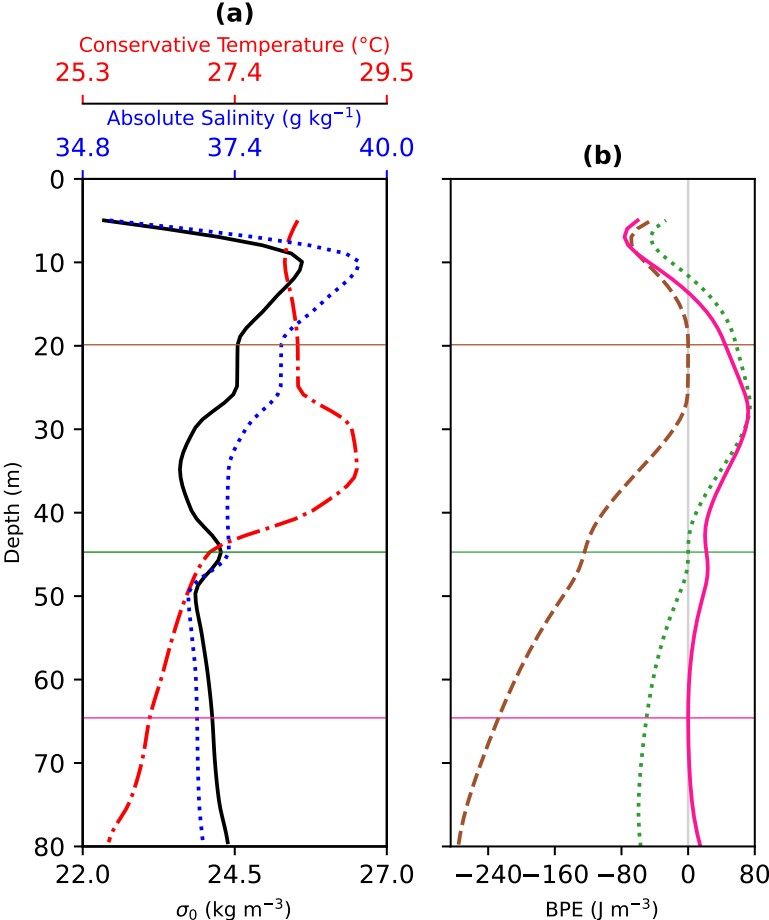

**Figure 2.** (a) Vertical profiles of potential density anomaly ($\sigma_0$) (black solid line), conservative temperature (red dash-dotted line), and absolute salinity (blue dotted line) for the AOML float number 4900755 on October 16, 2008, at [61.974°W, 19.505°N]. (b) BPE profiles considering equilibrium positions of 19.9 m (brown dashed line), 44.7 m (green dotted line), and 64.6 m (pink solid line).

maximum), or neutral (BPE is null). BPE also quantifies the energy required to displace the parcel from its equilibrium position to a certain depth: the more positive the BPE, the more energy is needed to displace the parcel; the more negative the BPE, the easier it is for the parcel to displace without additional energy. The magnitude of the energy barriers to the vertical displacement of a water parcel can serve as a proxy for the intensity of stratification experienced by a water parcel as it displaces vertically; the greater the BPE, the more stratified the water column, and vice versa. BPE is directly obtained from basic survey data without the need to diagnose the turbulent response of the water column (e.g., Franks, 2014; Sutherland et al., 2014; Reichl

et al., 2022). We provide a script that is straightforward to implement for calculating BPE from the potential density profile.

BPE is a proxy for stratification intensity that complements traditional stratification indexes, such as the potential energy anomaly (PEA) proposed by Simpson et al. (1978) and columnar buoyancy (CB) proposed by Herrmann et al. (2008). In





contrast to PEA and CB, BPE quantifies stratification intensity considering the direction in which it is measured (upward or
downward). Additionally, BPE has the advantage of providing a proxy for stratification intensity at any intermediate depth in
the water column, which is not present in prior stratification indexes that quantify upper stratification. Finally, note that the
square of the buoyancy frequency ($N^2$) cannot quantify stratification intensity along specific sections of the water column, as
it is a local measure of stratification.

## 3   Potential of BPE for enriching ocean analyses

Recent research has suggested that an adequate and relevant description of the water column stratification should be done in
terms of density, as this approach captures both temperature- and salinity-driven stratifications (Griffies et al., 2016; Sallée
et al., 2021; Treguier et al., 2023; Moreles et al., 2025). BPE is physically derived and density-based; therefore, it is well-
suited to measure stratification and has the potential to enrich ocean analyses. To illustrate that, this section presents a possible
roadmap for applying BPE to two example contexts: quantifying the intensity of barrier layers and describing the vertical
distribution of chlorophyll-$a$ (Chla).

### 3.1   Energy quantification of barrier layers

The barrier layers are the regions between the base of the density mixed layer and the top of the thermocline, which form
due to strong salinity stratification. They act as a barrier to heat and momentum exchange between the surface and the deep
ocean, thereby preventing vertical mixing and the turbulent entrainment of cold thermocline water into the surface mixed
layer (e.g., Sprintall and Tomczak, 1992; Cronin and McPhaden, 2002). Barrier layers are a crucial component in different
atmosphere-ocean interaction processes and the dynamics and thermodynamics of the ocean. For example, they constrain the
air-sea interactions within the mixed layer (e.g., Roemmich et al., 1994), may increase mixed layer warming (e.g., Echols
and Riser, 2020), affect the evaporation and convection rates from the ocean to the atmosphere (e.g., Ivanova et al., 2021),
could affect the vertical distribution of Chla concentration (e.g., Kawamiya and Oschlies, 2001), influence tropical cyclone
intensification by reducing sea surface temperature cooling (e.g., Wang et al., 2011; Balaguru et al., 2012; Vincent et al., 2012;
Yan et al., 2017), and affect El Niño-Southern Oscillation (e.g., Maes et al., 2005; Maes and Belamari, 2011; Guan et al., 2025).

To better understand the processes in which the barrier layers are relevant, it is crucial to characterize them accurately.
Historically, the barrier layers have been characterized by their thickness; however, this approach may have limitations. To
appreciate this, Fig. 3 shows examples of barrier layers in the Atlantic, Pacific, and Southern Oceans. We used the methodology
described by de Boyer Montégut et al. (2007) to calculate the mixed layer depth (MLD), the isothermal layer depth (ILD), and
the barrier layer thickness (BLT). The MLD is the depth at which the potential density increment from 10 m depth is equivalent
to a temperature decrease of 0.2°C at constant salinity. The ILD is the depth at which the temperature has decreased by 0.2°C
compared to its value at a depth of 10 m. Then, BLT = ILD - MLD. The BLT is 184 m in the Atlantic profile, 1120 m in the
Pacific profile, and 1123 m in the Southern profile.





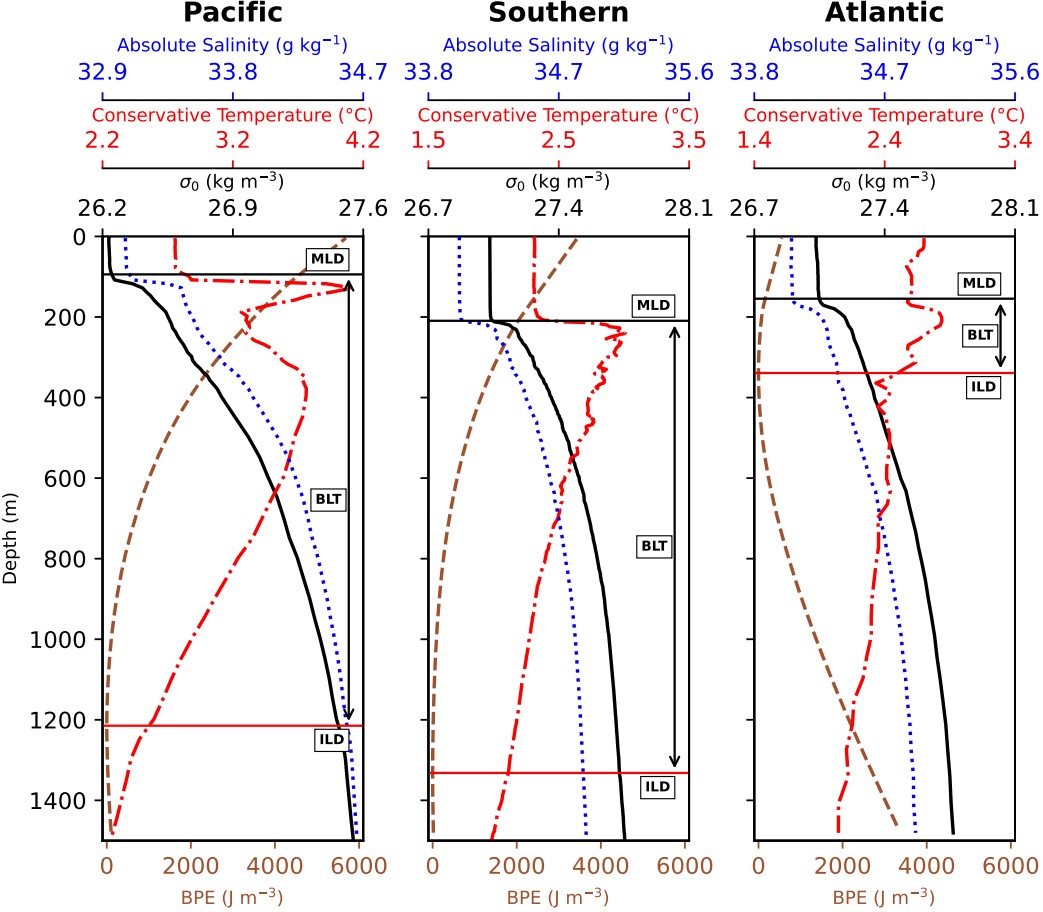

**Figure 3.** Profiles of barrier layers measured in the Pacific (179.899°W, 60.288°N on January 19, 2011), Southern (178.228°W, 60.904°S on June 25, 2015), and Atlantic (20.482°W, 51.418°S on October 6, 2009) Oceans. Profiles of potential density anomaly ($\sigma_0$) in black solid lines, conservative temperature in red dash-dotted lines, and absolute salinity in blue dotted lines. The BPE profiles (brown dashed lines) consider the ILD as the equilibrium position; the stratification intensity of the barrier layer is determined by the BPE value at the MLD.

Due to the strong salinity stratification and temperature inversion in the Pacific profile compared to the Southern profile, it is clear that the resistance to entrainment cooling, vertical mixing, and vertical exchanges of heat and momentum within the barrier layer is greater in the Pacific profile than in the Southern profile. However, the BLT is almost the same for those profiles (they differ by 3 m). It is concluded that the BLT is unable to account for the evident differences in the dynamic and thermodynamic characteristics of those barrier layers and for the intensity of the processes mentioned above occurring within

them. As pointed out by Reeves Eyre et al. (2019), the insulation effect of the barrier layers has not been entirely reviewed in the literature and remains an open question. The above demonstrates the need to characterize the barrier layers not only using the BLT but also in terms of their stratification intensity. We propose to add the stratification intensity along the barrier layers



(measured using BPE) as a complementary variable to characterize them. Given that the barrier layer is located between the MLD and ILD, the stratification intensity of the barrier layer can be measured by the BPE required to displace a water parcel

from the ILD to the MLD.

The application of BPE to characterize the barrier layers is also illustrated in Fig. 3. It shows the BPE profiles considering the ILD as the reference depth; in this way, the stratification intensity of the barrier layer is determined by the BPE value at the MLD. The stratification intensity is $160 \, \text{J} \cdot \text{m}^{-3}$ in the Atlantic profile, $4591 \, \text{J} \cdot \text{m}^{-3}$ in the Pacific profile, and $2033 \, \text{J} \cdot \text{m}^{-3}$ in the Southern profile. Although the BLT is almost the same for the Pacific and Southern profiles (they differ by 3 m),

their stratification intensity differs by $2558 \, \text{J} \cdot \text{m}^{-3}$. BPE can account for the differences in the dynamic and thermodynamic characteristics of these barrier layers, whereas the BLT does not. In this way, BPE complements traditional analyses of the ocean's dynamics and thermodynamics, allowing us to advance in their description.

The relationships between the BLT, entrainment cooling, and vertical diffusion into the mixed layer (e.g., Pham and Sarkar, 2017; Katsura et al., 2022) can be further enriched by exploring them in terms of the stratification intensity of the barrier

layer, as measured using BPE. The occurrence of barrier layers is commonly associated with shallow mixed layers, resulting in the trapping of surface fluxes within the mixed layer (e.g., Roemmich et al., 1994; Vialard and Delecluse, 1998a; Cronin and McPhaden, 2002). We can explore significant correlations between the sensitivity of the mixed layer to surface heat, momentum flux, and freshwater, in terms of the stratification intensity of the barrier layer measured using BPE (not only in terms of the BLT). An objective quantification of the relative stratification between the mixed layer and the barrier layer can help better

estimate the heat transfer across the isothermal layer base (e.g., Liu et al., 2022), thereby improving the understanding of the impact of barrier layers on El Niño development. More descriptive analyses of the spatiotemporal variability in the occurrence and intensity of barrier layers can be conducted by incorporating BPE into the analyses, thereby better characterizing the potential impact of barrier layers on the tropical cyclone intensification associated with a reduced sea surface temperature cooling (e.g., Wang et al., 2011; Balaguru et al., 2012; Vincent et al., 2012; Yan et al., 2017). We consider that incorporating

the stratification intensity of the barrier layer, as measured using BPE, can extend and enrich the analyses described above, potentially advancing our understanding of the dynamics and thermodynamics of barrier layers and their impact on diverse processes. Performing such analyses is beyond the scope of this study and is proposed for future research.

### 3.2   Refining the description of the vertical distribution of chlorophyll-$a$

In various oceanic and limnologic analyses, it is common to explain the vertical distribution of ecological (e.g., chlorophyll)

and chemical (e.g., dissolved oxygen) variables in terms of the mixed layer depth (MLD) and vertical stratification (e.g., Carvalho et al., 2017; Briseño-Avena et al., 2020; Gray et al., 2020; Cornec et al., 2021; Zampollo et al., 2023). However, there is high uncertainty in estimating the MLD (e.g., Gray et al., 2020; Tang et al., 2025; Moreles et al., 2025), which may lead to incomplete, inadequate, or inaccurate descriptions and predictions of ecological and chemical variables in relation to the MLD. To illustrate the above, Fig. 4 presents vertical profiles of Chla content and column-integrated Chla (CI-Chla) in the

Gulf of Mexico and a coastal sea in Antarctica, along with the MLD as estimated by five different methodologies. CI-Chla was calculated as the cumulative integral of Chla from its shallowest to its deepest record. The five MLD methodologies used are:



the $0.03 \ \mathrm{kg \cdot m^{-3}}$ and the 0.2°C thresholds of de Boyer Montégut et al. (2004), the multi-criteria method of Holte and Talley (2009), the maximum angle method of Chu and Fan (2011), and the sigmoid function fitting method of Romero et al. (2023). We refer to these methodologies as B04D, B04T, HT09, CF11, and R23, respectively. The BPE profiles consider the deepest

Deep Chla Maximum (DCM) as the equilibrium position.

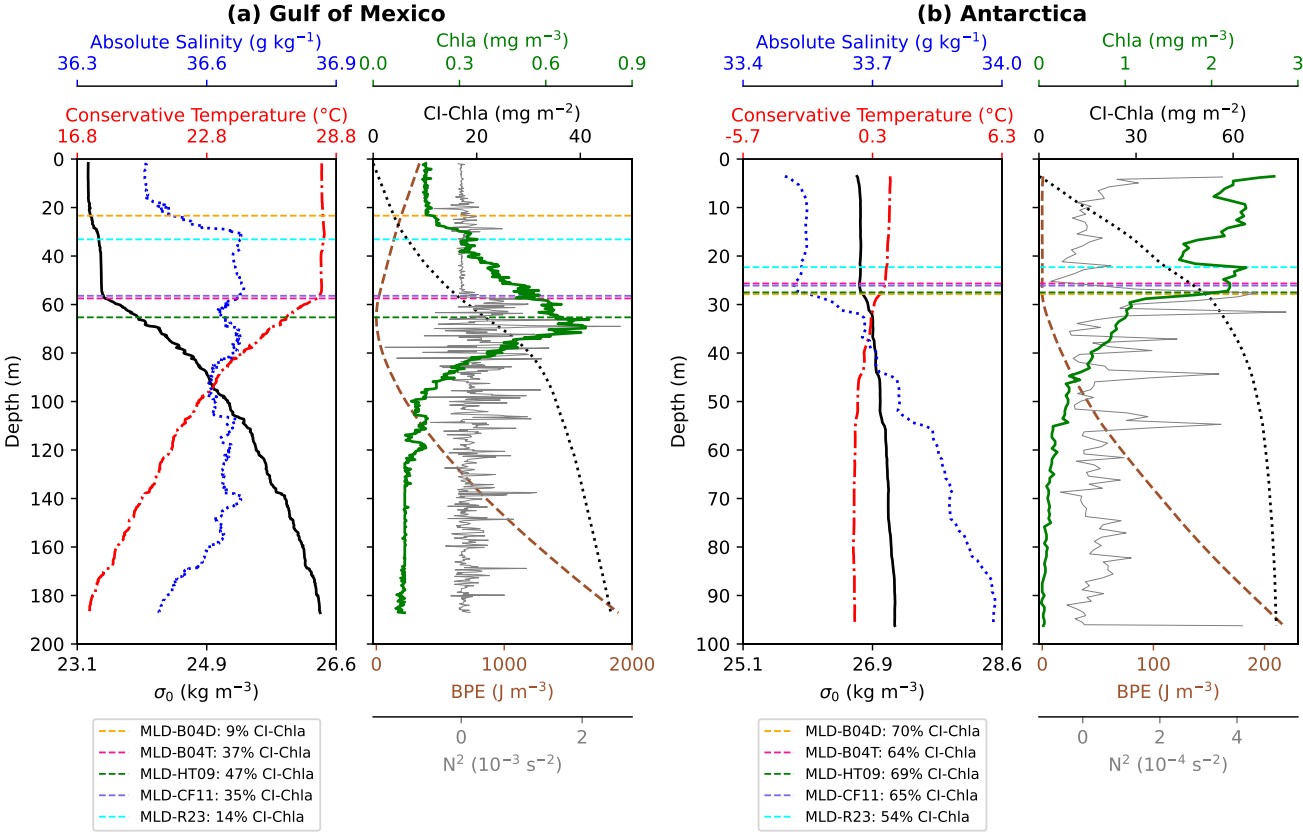

**Figure 4.** Vertical profiles of potential density anomaly ($\sigma_0$) (black solid line), conservative temperature (red dash-dotted line), absolute salinity (blue dotted line), Chla (green solid line), CI-Chla (black dotted line), BPE (brown dashed line), and $N^2$ (gray solid line) in (a) the Gulf of Mexico (92.41°W, 21.90°N on November 8, 2015) and (b) Antarctica (68.84°W, 67.86°S on January 27, 2009). The BPE profiles consider the deepest DCM (66 m for the Gulf of Mexico and 22 m for Antarctica) as the equilibrium position. The CI-Chla percentage at different MLD estimates is also shown, considering that the percentage increases from the shallowest to the deepest record.

Given the distinct oligotrophic characteristics of the Gulf of Mexico and Antarctica, their Chla content and CI-Chla profiles are very different, and their description varies a lot depending on the chosen MLD methodology:

– In the Gulf of Mexico (Fig. 4a), the Chla content is low in shallow waters, with a DCM at a depth of 66 m, which decreases to a constant value at a depth of 100 m. The MLDs are located between the upper section, characterized by

quasi-constant and low Chla content, and the depth of the DCM; the MLDs have variations of up to 42 m. B04D locates



the shallowest MLD, whereas HT09 locates the deepest MLD. The different MLDs are associated with varying CI-Chla percentages, ranging from 9 to 47% (38% of variation).

– In Antarctica (Fig. 4b), the Chla content has its highest value at the surface, decreasing with depth, with two DCMs at 10 and 22 m, followed by depletion at a depth of 100 m. All the MLDs are near the deepest DCM, with variations of up 190 to 6 m. However, the CI-Chla percentages can vary up to 16% (from 54 to 70%) between the MLDs.

Describing or predicting Chla content and CI-Chla in terms of the MLD could yield dissimilar results depending on the chosen MLD methodology (Fig. 4); the ambiguity in defining the MLD strongly limits its ability to explain and predict Chla content and CI-Chla. Further analyses based on an inadequate relationship between Chla content and stratification could lead to inaccurate or misleading assessments of phytoplankton biomass and descriptions of the nutrients required for phytoplank-195 ton growth. Recent research has focused on identifying significant relationships between the vertical distribution of diverse biochemical variables and the water column stratification (e.g., Vernet et al., 2008; Greer et al., 2013; Carvalho et al., 2017; Briseño-Avena et al., 2020; Gray et al., 2020; Cornec et al., 2021; Zampollo et al., 2023). The above emphasizes the need for an objective quantification of stratification, particularly at intermediate depths in the water column, to analyze the diverse ecological, biological, and chemical variables that may be affected by stratification. Since BPE provides an objective mea-200 sure of stratification intensity, we propose using this metric to complement the analyses of the vertical distribution of diverse biochemical variables in aquatic systems, to analyze how they vary across space and time, and to provide insight into their implications for ocean life processes. Our proposal for preferring BPE over the MLD does not underestimate the recent advances in accurately defining the MLD (e.g., Reichl et al., 2022; Moreles et al., 2025). Instead, BPE complements traditional analyses and can enhance our understanding of aquatic systems by identifying significant relationships between stratification and the 205 vertical distribution of diverse biochemical variables.

For a given profile of a biochemical variable, we can utilize its associated density profile to perform a BPE analysis and quantify stratification at different depths of biochemical interest, such as subsurface maxima or specific concentrations of the variable, or the occurrence of particular processes. From the deepest DCM in the Gulf of Mexico and Antarctica profiles, the stratification intensity differs between the two regions, both upward and downward (Fig. 4). From the DCM to the shallowest 210 record, the stratification intensity is $337 \, \mathrm{J \cdot m^{-3}}$ in the Gulf of Mexico and $1 \, \mathrm{J \cdot m^{-3}}$ in Antarctica; in contrast, the stratification intensity from the DCM to the deepest record is $1893 \, \mathrm{J \cdot m^{-3}}$ and $217 \, \mathrm{J \cdot m^{-3}}$, respectively, which undoubtedly impacts their primary production regimes. In the Gulf of Mexico profile (Fig. 4a), BPE reveals high energy barriers towards the surface and deep ocean, suggesting that the DCM may be retained by energy (stratification) barriers that limit vertical movement in both directions. In contrast, the Antarctic profile (Fig. 4b) exhibits a significant energy barrier only towards the deep ocean. 215 This asymmetry suggests an increased possibility for vertical dispersion (mixing) of Chla towards the surface. This exam-ple illustrates how BPE can complement the description of the vertical distribution of Chla content, thereby enhancing the characterization of the associated primary production regimes and biological activity in each region.

For instance, Carvalho et al. (2017) found that the ecologically relevant MLD is defined by the depth of the maximum value of $N^2$, as this depth represents a physical barrier that limits the mixing of phytoplankton in Antarctica's coastal seas. For the





profiles shown in Fig. 4, the depth of the maximum value of $N^2$ corresponds to the DCM in the Gulf of Mexico and to the depth at which Chla starts to deplete in Antarctica; the criterion found by Carvalho et al. (2017) does not always correspond to the same feature in the Chla content. The local nature of $N^2$ can be complemented by the vertically integrated measure of the stratification intensity provided by BPE, as described above.

## 4   Discussion and conclusions

This work proposes a metric for quantifying energy barriers associated with density stratification in vertical displacements of water parcels, which can be upward or downward. This energy-based metric, BPE, is useful for quantifying integrated stratification along any intermediate depth of the water column in terms of the energy barriers a water parcel would encounter during its upward and downward displacements. The above is not feasible when using local (e.g., the buoyancy frequency) or other integrated (e.g., Simpson et al., 1978; Herrmann et al., 2008) stratification metrics. A practical feature of BPE is that it is

directly calculated from survey data without the need to diagnose the turbulent response of the water column, and provides a direct and objective estimation of stratification intensity. The significance of BPE lies in its ability to provide a complementary understanding of oceanic stratification, thereby enriching diverse ocean analyses; moreover, it has a straightforward numerical implementation.

The approach to studying stratification in terms of energy and a scalar potential function is derived from classical physics.

Thus, that theory's physical and mathematical developments can equally apply to the study of stratification as proposed in this work, considering the physical setting underlying our approach (a fluid in hydrostatic balance). For instance, we can estimate to the very first order the speed a water parcel would require to displace a certain vertical distance, which is relevant in upwelling and downwelling processes (e.g., Rao et al., 2008), vertical migration of phytoplankton (e.g., Wirtz and Smith, 2020), and marine snow dynamics (e.g., Prairie et al., 2013). BPE could be used to revisit and refine relationships between dynamic and

ecological variables, such as those between stratification and Chla content (e.g., Carvalho et al., 2017; Briseño-Avena et al., 2020; Cornec et al., 2021), and vertical distribution of fish larvae (e.g., Sánchez-Velasco et al., 2007; Rodriguez et al., 2025); studies of that nature could be carried out in other regions and considering different time scales.

BPE can be used to refine and advance the understanding of the different air-sea interaction processes in which the barrier layers are relevant, spanning local to regional scales and various timescales (e.g., Vialard and Delecluse, 1998b; Masson et al.,

2004; Balaguru et al., 2012), for instance, the dynamics and thermodynamics of tropical cyclones. By investigating the dynamic connection between the thermal energy available to a tropical cyclone and stratification in the upper and intermediate ocean, we can gain deeper insights into the role of barrier layers in tropical cyclone intensification. The above is a critical issue given the increased hurricane intensity due to climate change (e.g., Holland and Bruyère, 2014). Furthermore, we can use this new approach to investigate physical associations between various ocean circulation processes and stratification, such as the Loop

Current intrusion and its eddy detachments in the Gulf of Mexico (e.g., Moreles et al., 2021; Higuera-Parra et al., 2023), upwelling in marginal seas (e.g., Reyes-Jiménez et al., 2023), spatial and temporal variability of ocean heat content (e.g., Buckley et al., 2019), ocean ventilation (e.g., Shepherd et al., 2017), and global trends in stratification (e.g., Li et al., 2020).



It is worth mentioning that dynamic equations, such as those developed by de Boer et al. (2008) and Burchard and Hofmeister (2008), offer a detailed analysis of the various processes that affect the evolution of stratification. However, these developments are intended for applications other than those of BPE, which represents a direct approach to quantify oceanic stratification. As mentioned in the derivation of BPE, it shares the same attributes as WB, but also exhibits the same downsides (Moreles et al., 2025). The most significant one is that BPE does not address active mixing and high-frequency variability, which are relevant when considering processes driven by synoptic atmospheric forcing, ocean eddies, and fronts. Additionally, as an integrated measure of stratification, BPE cannot describe local variations in stratification as $N^2$ does. It is pertinent to mention that BPE is not intended to replace prior stratification indexes (e.g., PEA, CB, and N2); instead, BPE complements them. The routine application of BPE to the study of different ocean-atmosphere interaction processes in which stratification plays a role can enhance our understanding of them.

*Code availability.* The methodology presented in this study is licensed under a GNU General Public License version 3.0. The source code is available at https://doi.org/10.5281/zenodo.14911394; the latest package version is v1.1.

*Author contributions.* EM conceived and designed the study, developed the methodology, wrote the scripts, interpreted the results, wrote and revised the manuscript, and acquired funding. ER designed the study, wrote the scripts, analyzed the data, interpreted the results, created the figures, and wrote and revised the manuscript. BM-L designed the study, interpreted the results, and revised the manuscript. All authors contributed substantially to this work and approved it for publication.

*Competing interests.* The authors declare that they have no conflict of interest.

*Acknowledgements.* The authors thank León Felipe Álvarez-Sánchez for contributing to preparing the scripts that calculate BPE and the generation of their repository on the GitHub platform. The authors used the writing assistant software Grammarly to review the manuscript's spelling and correct grammatical errors. This work was funded by UNAM-PAPIIT IN110925 and supported by the Instituto de Ciencias del Mar y Limnología of the Universidad Nacional Autónoma de México (grant 626).



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
