# Peer review of "Quantifying energy barriers associated with density stratification in vertical displacements of water parcels"

_EGUsphere, 2025_

## Referee Comment (RC1)

**Review of the Ocean Science manuscript egusphere-2025-3359**

Authors: Moreles, Romero and Martinez-Lopez

Reviewer: Trevor J McDougall, 28 July 2025

**Summary**. This manuscript is based on an incorrect expression
for the buoyant restoring force on a displaced fluid parcel. I
describe what I think is the correct approach, relying heavily on
Archimedes (personal communication 213BCE), and then
suggest how this energy required to displace a fluid parcel can be
calculated using the existing numerical algorithms in the TEOS-
10 Gibbs Seawater Toolbox (McDougall and Barker, 2011).

**1. Archimedes and the force on a displaced fluid parcel**

Like the authors of the above manuscript, consider a vertical water column of the ocean, and concentrate on a water parcel at a certain depth whose Absolute Salinity [for convenience we omit the usual subscript A of $S_A$], Conservative Temperature and Absolute Pressure (in

Pa) are $(\tilde{S}, \tilde{\Theta}, \tilde{P})$. The Absolute Salinity and Conservative Temperature of the ocean water column can be regarded as a function of pressure down the cast, that is, as $S(P)$ and $\Theta(P)$.

The parcel $(\tilde{S}, \tilde{\Theta}, \tilde{P})$ is now enclosed in an insulating plastic bag and is moved slowly until it arrives at a different depth in the water column. Let's call this new depth "location 2"

where the pressure is $P_2$, and the parcel's properties there are $(\tilde{S}, \tilde{\Theta}, P_2)$, while the surrounding ocean's properties at this location are $(S(P_2), \Theta(P_2), P_2)$. Along the journey to get from the original location to "location 2" the general pressure is labelled $P'$, and the parcel has properties $(\tilde{S}, \tilde{\Theta}, P')$, while the surrounding ocean at this general pressure has properties

$(S(P'), \Theta(P'), P')$.

We want to find the expression for the net buoyant force experienced by our fluid parcel

$(\tilde{S}, \tilde{\Theta}, P')$ at the general pressure $P'$ due to it being surrounded by the ocean environment fluid

$(S(P'), \Theta(P'), P')$ which has different salinity and temperature to that of our plastic-bag- enclosed parcel at this pressure. The key result that is needed here was discovered by

Archimedes of Syracuse and published by him in 213BCE; yes, published two thousand two hundred and thirty-eight (2038) years ago!  Archimedes (213BCE) stated that a floating body in equilibrium displaces a certain volume of the fluid in which it floats whose weight is that of the floating body.  [If I could talk to Archimedes, I would say to him how very impressive it is that he was able to prove this important result eighteen centuries before Isaac Newton developed the calculus.]

To be specific, let the mass of the fluid in our insulting plastic bag be 1 kilogram.  At its original location, the volume of this one kilogram of seawater is $\hat{v}(\tilde{S}, \tilde{\Theta}, \tilde{P})$, and its density is the reciprocal of this, namely $\hat{\rho}(\tilde{S}, \tilde{\Theta}, \tilde{P})$.  These quantities can be calculated from the GSW

Oceanographic Toolbox as gsw_specvol(SA_tilda,CT_tilda,P_tilda) and gsw_rho(SA_tilda,CT_tilda,P_tilda) respectively.  Note that the over-hat symbol of $\hat{v}$ and $\hat{\rho}$

simply draws our attention to the fact that these algorithms expect to have Conservative

Temperature as its temperature input.

Having moved our insulated seawater parcel adiabatically and without exchange of matter from $\tilde{P}$ to $P'$, the volume of this 1kg of seawater has changed from $\hat{v}(\tilde{S}, \tilde{\Theta}, \tilde{P})$ to $\hat{v}(\tilde{S}, \tilde{\Theta}, P')$.

This is the volume of seawater at $P'$ that our special bag displaces there.  Invoking the ground-breaking result of Archimedes (213BCE), the surrounding seawater provides an upwards force on our specially insulated fluid parcel of exactly the weight of seawater displaced.  What is this upwards force?  It is

Upwards force on the plastic bag $= g\ \hat{v}(\tilde{S}, \tilde{\Theta}, P')\ \hat{\rho}(S(P'), \Theta(P'), P')$.          (1)

This is understood as follows.  $\hat{\rho}(S(P'), \Theta(P'), P')$ is the mass per unit volume of the oceanic environment water at this pressure.  The mass of displaced oceanic water is the product of this with the volume of the displaced water, $\hat{v}(\tilde{S}, \tilde{\Theta}, P')$, and to convert this mass of displaced ocean water to a force, we multiply by the gravitational acceleration, $g$.

OK, we have the force that the ocean exerts on our bag-enclosed special parcel.  Now we need the weight of our parcel itself.  This is $g$.  That is, the gravitational acceleration times 1

kg, making 1 Newton.  If you like, this can also be found by a similar calculation to that of

Eq. (1), namely as $g\ \hat{v}(\tilde{S}, \tilde{\Theta}, P')\ \hat{\rho}(\tilde{S}, \tilde{\Theta}, P')$ which is simply $g$ since density and specific volume are reciprocals of each other.  So the net upwards vertical force acting on the parcel is

Net upwards force on the plastic bag $= g\ \{\hat{v}[\tilde{S}, \tilde{\Theta}, P']/\hat{v}[S(P'), \Theta(P'), P'] - 1\}$.      (2)

Thank you Mr Archimedes of Syracuse; we couldn't have done this without your insight.

**2. The work done to displace the fluid parcel**

Eq. (2) is the net force on the insulated bag containing our 1 kg of reference seawater. In order to keep the insulated bag at this location we have to supply an equal and opposite external force to it [hence the minus sign at the beginning of the right-hand side of Eq. (3) below]. This force grows from the reference location, and we wish to quantify the total energy required to slowly move the insulated bag of fluid from its location at $\tilde{P}$ to the final location at $P_2$. This energy is the integral of the force with respect to height,

$$\text{Energy required per kg } = -\int_{\tilde{z}}^{z_2} g \left\{ \frac{\hat{v}[\tilde{S}, \tilde{\Theta}, P']}{\hat{v}[S(P'), \Theta(P'), P']} - 1 \right\} dz'. \tag{3}$$

Now we use the hydrostatic relationship, $P_z = -g\rho$, or $g\,z_P = -\hat{v}[S(P'), \Theta(P'), P']$, to find

$$\text{Energy required per kg } = \int_{\tilde{P}}^{P_2} \left\{ \hat{v}[\tilde{S}, \tilde{\Theta}, P'] - \hat{v}[S(P'), \Theta(P'), P'] \right\} dP'. \tag{4}$$

This is the accurate version of equation (3) of the submitted manuscript 2025-3359.

**3. Evaluating the work done using GSW software**

Eq. (4) above is almost identical to the Cunningham geostrophic streamfunction (see section 3.29 of the TEOS-10 Manual, IOC et al. 2010). This means that the present submitted manuscript egusphere-2025-3359 provides a nice physical explanation of the Cunnigham geostrophic streamfunction; an explanation/justification that is new to this reviewer.

The difference between Eq. (4) above and Eqs. (3.19.1) and (3.19.2) of the TEOS-10 Manual is that there the Cunningham geostrophic streamfunction is set up with the reference pressure being the sea surface pressure $P_0$ rather than being a general starting pressure $\tilde{P}$ as we do here. Hence, I will use the gsw software, `gsw_geo_strf_dyn_height(SA,CT,p,p_ref)` as the basis for writing Eq. (4) above in terms of GSW algorithms. This `gsw_geo_strf_dyn_height` algorithm delivers Eq. (3.27.1) of the TEOS-10 Manual. In terms of this function, we can evaluate Eq. (4) above by

$\text{Energy required per kg} \; = \; \int_{\tilde{P}}^{P_2} \{\hat{v}[\tilde{S}, \tilde{\Theta}, P'] - \hat{v}[S(P'), \Theta(P'), P']\} \, dP'$

$= \; \text{gsw\_enthalpy(SA\_tilda, CT\_tilda, p\_2)}$

$- \; \text{gsw\_enthalpy(SA\_tilda, CT\_tilda, p\_tilda)}$

$+ \; \text{gsw\_geo\_strf\_dyn\_height(SA,CT, p, p\_tilda)}$ (5)

$- \; \text{gsw\_enthalpy(SSO, 0°C, p\_2)}$

$+ \; \text{gsw\_enthalpy(SSO, 0°C, p\_tilda)}.$

**4. Remarks on the approximate approach based on potential density**

Equation (1) of the submitted manuscript 2025-3359 is an approximate expression for the net force on our insulating impervious bag. This approximate expression ignores an important nonlinearity of the equation of state: the thermobaric nonlinearity. To see this, note that the manuscript's approximate expression for the force per unit mass is

$\text{Approximate net force} \; = \; g \left\{ \hat{v}[\tilde{S}, \tilde{\Theta}, P_2] / \hat{v}[S(P'), \Theta(P'), P_2] - 1 \right\},$ (6)

[where the two specific volumes are the potential specific volumes refereed to the reference pressure $P_2$] which is to be compared with the accurate net force of Eq. (2) above. In the approximate case the force is proportional to $\{\hat{\rho}[S(P'), \Theta(P'), P_2] - \hat{\rho}[\tilde{S}, \tilde{\Theta}, P_2]\}$ while in the accurate case it is proportional to $\{\hat{\rho}[S(P'), \Theta(P'), P'] - \hat{\rho}[\tilde{S}, \tilde{\Theta}, P']\}$. The (linearized)

difference between these two expressions is

$\text{Error in net force per kg} \; \approx \; \{P_2 - P'\} \left\{ \hat{\rho}_P \left[S(P'), \Theta(P'), \frac{(P'+P_2)}{2}\right] - \hat{\rho}_P \left[\tilde{S}, \tilde{\Theta}, \frac{(P'+P_2)}{2}\right] \right\}.$ (7)

Concentrating on the dependence of the adiabatic compressibility, on temperature (and ignoring its dependence on salinity) this Eq. (7) scales as

$\text{Error in net force per kg} \; \approx \; \{P_2 - P'\} \{\Theta(P') - \tilde{\Theta}\} \hat{\rho}_{P\Theta}.$ (8)

Eq. (8) contains the product of a pressure difference and a temperature difference, multiplied by $\hat{\rho}_{P\Theta}$ which represents the rate at which the adiabatic compressibility (square of sound speed) varies with temperature, or equivalently, the rate at which the thermal expansion coefficient varies with pressure. This is the essence of the thermobaric non-linearity of the equation of state.

As with all uses of potential density, what is ignored is related to the thermobaric non-
linearity of the equation of state of seawater. That is, the approach of Geoff Vallis' textbook
is approximate, and it ignores this thermobaric nonlinearity. For large vertical excursions,
this approximation can result in huge errors. As an example, consider that potential density
referenced to the sea surface actually **decreases** with pressure in the deep North Atlantic,
even though the water column there is statically stable. Deploying the method of this draft
manuscript 2025-3359 would be greatly in error in such a region.

The thermobaric nonlinearity of the equation of state of seawater is actually sufficiently
large that a salinity-temperature diagram as below is theoretically possible.

[Figure]

This SA-CT diagram is of a single vertical CTD cast which doubles back on itself but is
statically stable at all heights. An analysis based on potential density would be hopelessly
inaccurate when applied to such an extreme SA_CT vertical cast.

**5. Concluding remarks**

When the manuscript is corrected so that the expression for the net buoyant force is
correct at finite amplitude, is it publishable as a new contribution to oceanographic
knowledge? Perhaps it is. I, for one, did not know the connection between the Cunningham
geostrophic streamfunction and the energy required to move an insulated parcel through the
range of pressures. This connection between gravitational potential energy and a geostrophic
streamfunction was new to me and only revealed itself when carefully deriving the above
results, using the key insightful result of Archemedes of Syracuse.

REFERENCES

Archimedes, 216BC: On Floating Bodies I, *translated and reproduced in full in*

Heath, T. L., 1897: The Works of Archimedes, Cambridge University Press, Cambridge,

326pp.  Available at https://dn790000.ca.archive.org/0/items/worksofarchimede00arch/worksofarchimede00arch.pdf

IOC, SCOR and IAPSO, 2010: *The international thermodynamic equation of seawater – 2010:*

*Calculation and use of thermodynamic properties*.  Intergovernmental Oceanographic

Commission, Manuals and Guides No. 56, UNESCO (English), 196 pp.  Available from www.TEOS-10.org [IOC et al. (2010) is colloquially called "the TEOS-10 manual"]

McDougall T. J. and P. M. Barker, 2011: Getting started with TEOS-10 and the Gibbs

Seawater (GSW) Oceanographic Toolbox, 28pp., SCOR/IAPSO WG127, ISBN 978-0-

646-55621-5, available from www.TEOS-10.org

---

## Author Comment (AC1)

**Discussion of the manuscript: <a href="https://doi.org/10.5194/egusphere-2025-3359">https://doi.org/10.5194/egusphere-2025-3359</a>**

**Title: Quantifying energy barriers associated with density stratification in vertical displacements of water parcels**

Dear Professor McDougall. We appreciate the time you take to review our preprint, the detailed mathematical derivation of the buoyant force, and the pertinent comments regarding our approach. As you suggested, the analysis regarding the net force on a displaced fluid parcel was carefully revised and corrected; we provide a better contextualization of our formulation and its limitations. Also, we revised the Introduction, incorporated recent relevant literature, and clarified the objective of our study. We hope you find the corrections satisfactory and that our research contributes to the advancement of oceanographic knowledge.

**Reviewer #1 Trevor J McDougall**

**Comment 1:** This manuscript is based on an incorrect expression for the buoyant restoring force on a displaced fluid parcel. I describe what I think is the correct approach, relying heavily on Archimedes (personal communication 213BCE), and then suggest how this energy required to displace a fluid parcel can be calculated using the existing numerical algorithms in the TEOS-10 Gibbs Seawater Toolbox (McDougall and Barker, 2011).

When the manuscript is corrected so that the expression for the net buoyant force is correct at finite amplitude, is it publishable as a new contribution to oceanographic knowledge? Perhaps it is. I, for one, did not know the connection between the Cunningham geostrophic streamfunction and the energy required to move an insulated parcel through the range of pressures. This connection between gravitational potential energy and a geostrophic streamfunction was new to me and only revealed itself when carefully deriving the above results, using the key insightful result of Archimedes of Syracuse.

**Answer 1:** We appreciate the time you take to review our manuscript, the detailed mathematical derivation of the buoyant force, and the pertinent comments regarding our approach. We agree with you that the expression to calculate the net force is incorrect; it comes from an approximation to the force in terms of the potential density of the environment.

In the revised version of the methodology (see the attached document), we revised our calculations and corrected the expression of the force by describing the physical setting of the problem and the different approaches we took to obtain the final expression of the force (which is an approximated expression). Using real ocean profiles, we examined the accuracy of the approximated expression for the force and found that calculating the force with the potential density of the environment referenced to a fixed pressure, centered at the section of interest, is sufficient for qualitative oceanography. The integrated error (measured in terms of the mean absolute percentage error) is less than 5% for pressure variations not exceeding 2000 dbar.

From the corrected expression for the force, we were able to define the buoyancy potential energy (BPE) to estimate the energy barriers associated with density stratification in vertical displacements of water parcels. We also calculated the energy required to vertically displace an insulated water parcel using the expression for BPE and the algorithms in the TEOS-10 toolbox suggested by you, considering real ocean profiles. As with the force, the integrated error at each depth (measured in terms of the mean absolute percentage error) is less than 5% in vertical sections with pressure variations of up to 2000 dbar.

The connection you revealed between BPE and the Cunningham geostrophic streamfunction is very interesting and worth analyzing in more detail in future work. We greatly appreciate this clever derivation, which will undoubtedly enhance our future research.

**1 1 Introduction**

The vertical variation in density in aquatic bodies (i.e., oceans, seas, freshwater bodies, and estuaries), resulting from temperature and salinity differences, constitutes their stratification, which has significant implications for the variability of weather and climate processes as it determines different dynamical, ecological, chemical, and biological processes occurring there. Stratification has a significant role on the flux of particulate organic matter from the surface to sediments (e.g., Kirillin et al., 2012; Omand et al., 2020), the vertical content of chlorophyll (e.g., Cullen, 2015; Carvalho et al., 2017; Briseño-Avena et al., 2020; Cornec et al., 2021; Zampollo et al., 2023), biological productivity (e.g., Franks, 2014; Bouman et al., 2020), the mixed layer (e.g., Sutherland et al., 2014; Gray et al., 2020; Moreles et al., 2025), the 10 ocean-atmosphere coupling and exchanges between them (e.g., Deser et al., 2010; Groeskamp 11 et al., 2019), barrier layers (e.g., Sprintall and Tomczak, 1992; Cronin and McPhaden, 2002), 12 and tropical cyclone dynamics (e.g., Wang et al., 2011; Balaguru et al., 2012; Vincent et al., 13 2012; Yan et al., 2017). 14

16

The vertical structure of aquatic bodies in terms of their stratification and energetics has been intensively studied. Schmidt (1928) derived an equation for the stability of a lake; Idso (1973) subsequently extended that equation to account for the contribution of each lake layer to stability. Simpson et al. (1978) and Herrmann et al. (2008) proposed energy-based indexes to quantify stratification from the surface to a specific depth. Dynamic equations derived from Simpson et al.'s index are used to analyze the evolution of stratification in terms of different processes (Simpson and Bowers, 1981; Burchard and Hofmeister, 2008; de Boer et al., 2008). Using boundary-layer turbulence theory, Reichl et al. (2022) proposed the potential energy anomaly of the water column to estimate the depth to which a given energy could homogenize a layer of seawater; from this, they introduced a framework for diagnosing the ocean mixed layer depth. The potential energy anomaly provides a proxy for the stratification of a layer of seawater by estimating its energetic distance from a well-mixed state. From energetic foundations, Rosenthal and Roquet (2025) developed an index of stratification strength for the global ocean using the height anomaly, defined as the height of the ocean's center of mass relative to the height of a fully-mixed state. The lower the center of mass, the larger the stratification, and vice versa. They also developed a tendency equation for the budget of the height anomaly, which helps identify the overall contribution of different processes to the stratification. Moreles et al. (2025) proposed the buoyancy work required to displace a water parcel vertically as a proxy for the vertical homogeneity of the water column: the lower the work, the greater the vertical homogeneity of the water column, and vice versa. From this, they defined the ocean mixed layer and computed a global monthly climatology of its depth that maintains quasi-homogeneity in energy, density, and temperature along the mixed layer year-round.

39 40

Stratification determines mixing, which is commonly associated with the interchange of properties between deep and shallow waters. This vertical interchange of properties is typically linked to the vertical displacement of water parcels, which carry their properties from one isopycnal to another. The vertical movement of a water parcel is, to a certain extent, determined by the density stratification experienced by the water parcel when it displaces upward and downward; the more stratified the water column, the more energy is required to displace the water parcel vertically, and vice versa. What are the characteristics of the energy barriers associated with density stratification in vertical displacements of water parcels? The above is particularly relevant when studying processes (physical, ecological, chemical, or biological) that occur in the subsurface or at a distance from the surface, where the direction of vertical movement of water parcels is relevant. For any intermediate section in the water column, the energy barriers within it depend on the direction (upward or downward) in which they are measured: the barriers experienced by a deep-water parcel when it displaces upward differ from those experienced by a shallow-water parcel when it displaces downward.

The above question has not been addressed in previous studies; thus, our objective is to estimate the energy barriers associated with density stratification in vertical displacements of water parcels. For a water parcel at any depth within the water column, we estimate the energy barriers it would encounter if vertically displaced upward or downward. The magnitude of the energy barriers to vertical displacement of a water parcel can serve as a proxy for the intensity of stratification it experiences along its vertical displacement; the greater the barriers, the more stratified the water column, and vice versa. By focusing on the direction of a water parcel's vertical displacement, we can analyze stratification in terms of the direction of quantification, thus enhancing the analyses of stratification in aquatic bodies. The metric used to estimate energy barriers is presented in Section 2, followed by its application to dynamical and ecological contexts in Section 3, which demonstrates its potential for enriching ocean analyses.

**67 2 Estimating energy barriers**

To analyze the energy barriers, we quantified the buoyancy energy required for a water parcel in equilibrium to be displaced upward and downward. Given a vertical density stratification, we first derive an expression to estimate the force F on a water parcel, initially at rest, if it were displaced vertically (upward or downward) from its equilibrium position. If the force is conservative (it is path-independent), we can use classical mechanics arguments to obtain a potential function V from which the force F can be derived (Goldstein, 1980, p. 4),

$$\vec{F} = -\nabla V. \tag{1}$$

The potential function V is the buoyancy energy and is a function of the vertical coordinate. It represents the energy barriers a water parcel would encounter along its vertical movement.

The physical setting of the problem involves a fluid in hydrostatic balance in a constant gravitational field, where rotation effects, horizontal motion, and friction are neglected. This physical setting is typically used when analyzing the oscillation of a fluid perturbed away from its resting state, and the Brunt-Väisälä frequency is derived. For our derivation, we will start from the expression for the net force; for a detailed description of this physical setting and the derivation of the net force, see sections 2.9.1 and 2.9.2 of Vallis (2006) and McDougall (2025). A water parcel at its equilibrium position  $z_{eq}$  that is slowly vertically displaced from that level to any depth z, without exchanging either mass or heat with the surroundings, experiences a net force per unit volume given by (Vallis, 2006, p. 92),

$$F(z) = g\left[\hat{\rho}(z) - \rho(z)\right],\tag{2}$$

where g is the acceleration due to gravity,  $\hat{\rho}$  is the in-situ density of the environment, and  $\rho$  is the in-situ density of the parcel.

Since the parcel is undergoing adiabatic conditions during the displacement, the net force (Eq. 2) can be expressed as a function of pressure P as (McDougall, 2025),

$$g\{\hat{\rho}[S(P), \Theta(P), P] - \hat{\rho}[S_{eq}, \Theta_{eq}, P]\}$$
 force per unit volume, (3a)

$$g\left\{\frac{\hat{v}\left[S_{\text{eq}},\Theta_{\text{eq}},P\right]}{\hat{v}\left[S(P),\Theta(P),P\right]}-1\right\} \quad \text{force per unit mass,}$$
 (3b)

where  $\hat{v}$  is the specific volume of the environment (the reciprocal of  $\hat{\rho}$ ) and the absolute salinity and conservative temperature of the environment have been regarded as a function of P as S(P) and  $\Theta(P)$ , respectively. The subscript eq refers to the properties of the parcel at the equilibrium position  $z_{\rm eq}$  or  $P_{\rm eq}$ .

Equations (3), derived from first principles, represent the accurate expressions to calculate the net force on the water parcel when it is vertically displaced from its equilibrium position  $z_{\text{eq}}$  (or  $P_{\text{eq}}$ ) to any depth z (or P) under adiabatic conditions. However, they are not easily structured to calculate the associated potential function using Eq. (1) because the force is a composite function of pressure in terms of absolute salinity and conservative temperature. To deal with this problem, we explored an approximation for the force, valid for small displacements, calculated in terms of the potential density of the environment referenced to the pressure at the level z (Vallis, 2006, p. 93),

$$F(z) \approx g \left[ \rho_{\theta}(z) - \rho_{\theta}(z_{\text{eq}}) \right],$$
 (4)

where  $\rho_{\theta}$  is the locally-referenced potential density. Eq. (4) is still insufficient to calculate the associated potential function due to the potential density is not referenced to a fixed pressure, which results in that the vertical coordinate is not unique throughout the displacement. In order to calculate the associated potential function using Eq. (1), we must calculate the force with Eq. (4) but using the potential density referenced to a fixed pressure. However, the use of potential density referenced to a fixed pressure ignores the thermobaric effect, which can lead to significant errors for large vertical displacements of the water parcel (McDougall,

1987a,b). Thus, it is necessary to examine whether this approximation of the force is accurate enough to replace the accurate expression (Eqs. 3), at least for qualitative oceanography.

We selected three Argo profiles (first column of Fig. 1), exhibiting different stratification conditions, to analyze the differences between various versions of the force: (i) the accurate expression given by Eq. (3b) and the approximate expressions given by Eq. (4), considering (ii) the locally-referenced potential density and (iii) the potential density referenced to a fixed pressure. For each profile, we calculated the forces at various depths, considering that the equilibrium position  $P_{\rm eq}$  of the water parcel is at the isothermal layer depth, defined as the depth at which the conservative temperature has decreased by 0.2°C from the temperature at a depth of 10 m (de Boyer Montégut et al., 2004). Then, we calculated the differences between the approximate forces and the accurate force. Finally, we calculated the mean absolute percentage error (MAPE) between each approximate force and the accurate one at various depths; from  $P_{\rm eq}$ , we selected a vertical section that increased in length (both upward and downward) to calculate the MAPE. We used the Thermodynamic Equation of SeaWater 2010 (McDougall and Barker, 2011) to calculate the different variables in the equations of interest.

A visual inspection of the results shows that the calculated force is very similar across all versions (second column of Fig. 1). At each depth, the differences between the approximate forces with respect to the exact force are three orders of magnitude smaller if the locally-referenced potential density is used and one order of magnitude smaller if the potential density is referenced to a fixed pressure (third column of Fig. 1). The differences at  $P_{\rm eq}$  are zero and they increase for depths far from  $P_{\rm eq}$  and far from the reference pressure used to calculate the potential density. However, since we are interested in integrated measures, the MAPE at various depths is a better measure to quantify the error of the approximate forces (fourth column of Fig. 1). The force calculated with Eq. (4) using the locally-referenced potential density is nearly the same as the accurate one, with MAPE of less than 1% throughout the vertical. The MAPE can be very large when using the potential density referenced to a fixed pressure far from  $P_{\rm eq}$ , even if the vertical displacements of the parcel are small; the MAPE can reach up to 30% when the differences between  $P_{\rm eq}$  and the reference pressure exceed 1500 dbar. When using Eq. (4) with the potential density referenced to  $P_{\rm eq}$ , the MAPE values are less than 5% throughout the vertical (with pressure variations of up to 2000 dbar).

Using inductive reasoning, we assume that the above results are maintained for the world ocean with pressure variations of 2000 dbar, suggesting the following. The approximate force (Eq. 4) calculated with the potential density referenced to a fixed pressure, centered in the section of interest, is sufficient for qualitative oceanography; the integrated error will presumably be less than 5% for pressure variations not exceeding 2000 dbar. This suggestion is in agreement with the findings of Lynn and Reid (1968) and Reid and Lynn (1971), who observed that if the vertical section of interest does not exceed pressure variations of about 1000 dbar, the stability of the water column is adequately described with the potential den-

Figure 1: **First column:** ocean profiles in the Pacific, Southern, and Atlantic Oceans; the profiles of potential density anomaly  $\sigma$  are referenced to  $P_{\rm eq}$ . **Second column:** accurate net force and approximations to it calculated with different versions of potential density. **Third column:** differences between the approximate forces and the accurate force. **Fourth column:** the MAPE of each approximate force at various depths, calculated from  $P_{\rm eq}$ .

sity referenced to a pressure centered in the section of interest. Despite the high accuracy expected when using Eq. (4), we can always calculate the accurate force via Eqs. (3) and quantify the integrated error associated with using the approximate expression.

The above justify using the approximate expression (Eq. 4) to calculate the net force on the water parcel when it is vertically displaced from its equilibrium position  $z_{eq}$  to any depth z. Since we are using the potential density referenced to a fixed pressure, the vertical coordinate is unique throughout the displacement and we can calculate the potential energy function associated with the net buoyant force, the buoyancy potential energy (BPE), using

$$F(z) = -\frac{\mathrm{d}}{\mathrm{d}z} \mathrm{BPE}(z). \tag{5}$$

BPE is obtained by vertically integrating the force in Eq. (4),

Eq. (1),

$$BPE(z) = BPE(z_{eq}) - \int_{z_{eq}}^{z} g \left[ \rho_{\theta}(\gamma) - \rho_{\theta}(z_{eq}) \right] d\gamma = BPE(z_{eq}) + g(z - z_{eq}) \rho_{\theta}(z_{eq}) - g \int_{z_{eq}}^{z} \rho_{\theta}(\gamma) d\gamma,$$
(6)

where  $\rho_{\theta}$  is the potential density of the environment referenced to a fixed pressure, centered in the section of interest. When working with potentials, the physically relevant quantity is the potential difference between two depths; thus, we can set BPE( $z_{eq}$ ) = 0 without loss of generality. BPE represents the energy barriers a water parcel would encounter if it were displaced from its equilibrium position  $z_{eq}$  to any depth z. The work done by the net buoyant force in displacing a water parcel from  $z_1$  to  $z_2$  is BPE( $z_1$ ) – BPE( $z_2$ ). BPE is directly related to the work done by buoyancy proposed by Moreles et al. (2025); thus, all the properties and attributes of the work done by buoyancy are directly applicable to BPE.

The expression given by Eq. (6) is an approximate expression for the total energy required to slowly move an insulated parcel of fluid from its equilibrium location to any final location. The accurate expression is given by vertically integrating the force in Eqs. (3), as shown by McDougall (2025) in his Eqs. (3) and (4). Through a meticulous and detailed review of this preprint, McDougall (2025) identified a connection between BPE and the Cunningham geostrophic streamfunction, a novel result. He then proposed a way to calculate this energy using the TEOS-10 Toolbox (see his Eq. 5). We computed the energy for each profile shown in Fig. 1 using BPE (Eq. 6) and Eq. 5 of McDougall (2025) (plots not shown). Similar to what we found in the force analysis, the differences in the energy values at each depth obtained with these two expressions are minimal (the MAPE between them is less than 5% throughout the vertical), suggesting that BE is accurate enough for calculating the energy in vertical sections not exceeding pressure variations of 2000 dbar. Again, we can always calculate the energy using the accurate expression and quantify the integrated error associated with using the approximate expression.

Our approach provides a physically derived, approximated, and intuitive variable (i.e.,

BPE) to estimate the energy barriers associated with density stratification in vertical displacements of water parcels, which is accurate enough for qualitative oceanography. Depending on the sign of BPE, two physical situations are identified. For BPE > 0, the force and the parcel displacement are in opposite directions, causing the parcel to decelerate when ... ... it continues as it is in the preprint from line 68.

Note: According to the new way for calculating BPE, Fig. 3 regarding the barrier layers will be adjusted to reflect the BPE calculated using the potential density referenced to a fixed pressure, centered between the mixed layer depth and the isothermal layer depth. The prior BPE and the new BPE are nearly identical; therefore, the discussion and results of the new figure are maintained as in the preprint version with this adjustment.

**References**

- K. Balaguru, P. Chang, R. Saravanan, L. R. Leung, Z. Xu, M. Li, and J.-S. Hsieh. Ocean barrier layers' effect on tropical cyclone intensification. *Proceedings of the National Academy of Sciences*, 109(36):14343–14347, 2012. doi: 10.1073/pnas.1201364109.
- H. A. Bouman, T. Jackson, S. Sathyendranath, and T. Platt. Vertical structure in chlorophyll profiles: influence on primary production in the Arctic Ocean. *Philosophical Transac*tions of the Royal Society A: Mathematical, Physical and Engineering Sciences, 378(2181): 20190351, 2020. doi: 10.1098/rsta.2019.0351.
- C. Briseño-Avena, J. C. Prairie, P. J. S. Franks, and J. S. Jaffe. Comparing Vertical Distributions of Chl-a Fluorescence, Marine Snow, and Taxon-Specific Zooplankton in Relation to Density Using High-Resolution Optical Measurements. Frontiers in Marine Science, 7, 2020. ISSN 2296-7745. doi: 10.3389/fmars.2020.00602.
- H. Burchard and R. Hofmeister. A dynamic equation for the potential energy anomaly for analysing mixing and stratification in estuaries and coastal seas. *Estuarine*, *Coastal and Shelf Science*, 77(4):679–687, 2008. ISSN 0272-7714. doi: 10.1016/j.ecss.2007.10.025.
- F. Carvalho, J. Kohut, M. J. Oliver, and O. Schofield. Defining the ecologically relevant mixed-layer depth for Antarctica's coastal seas. *Geophysical Research Letters*, 44(1):338–345, 2017. doi: 10.1002/2016GL071205.
- M. Cornec, H. Claustre, A. Mignot, L. Guidi, L. Lacour, A. Poteau, F. D'Ortenzio, B. Gentili,
   and C. Schmechtig. Deep Chlorophyll Maxima in the Global Ocean: Occurrences, Drivers
   and Characteristics. Global Biogeochemical Cycles, 35(4):e2020GB006759, 2021. doi: 10.
   1029/2020GB006759. e2020GB006759 2020GB006759.
- M. F. Cronin and M. J. McPhaden. Barrier layer formation during westerly wind bursts.

  Journal of Geophysical Research: Oceans, 107(C12):SRF 21–1–SRF 21–12, 2002. doi: 10.1029/2001JC001171.

- J. J. Cullen. Subsurface Chlorophyll Maximum Layers: Enduring Enigma or Mystery Solved?
   Annual Review of Marine Science, 7(Volume 7, 2015):207–239, 2015. ISSN 1941-0611. doi:
   10.1146/annurev-marine-010213-135111.
- G. J. de Boer, J. D. Pietrzak, and J. C. Winterwerp. Using the potential energy anomaly equation to investigate tidal straining and advection of stratification in a region of freshwater influence. *Ocean Modelling*, 22(1):1–11, 2008. ISSN 1463-5003. doi: 10.1016/j.ocemod.2007.12.003.
- C. de Boyer Montégut, G. Madec, A. S. Fischer, A. Lazar, and D. Iudicone. Mixed layer depth over the global ocean: An examination of profile data and a profile-based climatology.
   Journal of Geophysical Research: Oceans, 109(C12), 2004. doi: 10.1029/2004JC002378.
- C. Deser, M. A. Alexander, S.-P. Xie, and A. S. Phillips. Sea Surface Temperature Variability:
   Patterns and Mechanisms. Annual Review of Marine Science, 2(Volume 2, 2010):115–143,
   2010. ISSN 1941-0611. doi: 10.1146/annurev-marine-120408-151453.
- P. J. S. Franks. Has Sverdrup's critical depth hypothesis been tested? Mixed layers vs. turbulent layers. *ICES Journal of Marine Science*, 72(6):1897–1907, 10 2014. ISSN 1054-3139. doi: 10.1093/icesjms/fsu175.
- H. Goldstein. *Classical Mechanics*. Addison-Wesley series in physics. Addison-Wesley, Reading, Mass., 2nd ed edition, 1980.
- E. Gray, E. B. Mackay, J. A. Elliott, A. M. Folkard, and I. D. Jones. Wide-spread inconsistency in estimation of lake mixed depth impacts interpretation of limnological processes.

  Water Research, 168:115136, 2020. ISSN 0043-1354. doi: 10.1016/j.watres.2019.115136.
- S. Groeskamp, S. M. Griffies, D. Iudicone, R. Marsh, A. G. Nurser, and J. D. Zika. The Water Mass Transformation Framework for Ocean Physics and Biogeochemistry. *Annual Review of Marine Science*, 11(Volume 11, 2019):271–305, 2019. ISSN 1941-0611. doi: 10.1146/annurev-marine-010318-095421.
- M. Herrmann, S. Somot, F. Sevault, C. Estournel, and M. Déqué. Modeling the deep convection in the northwestern Mediterranean Sea using an eddy-permitting and an eddy-resolving model: Case study of winter 1986–1987. *Journal of Geophysical Research:*Oceans, 113(C4), 2008. doi: 10.1029/2006JC003991.
- S. B. Idso. On the concept of lake stability. *Limnology and Oceanography*, 18(4):681–683, 1973. doi: 10.4319/lo.1973.18.4.0681.
- G. Kirillin, H.-P. Grossart, and K. W. Tang. Modeling sinking rate of zooplankton carcasses:
   Effects of stratification and mixing. Limnology and Oceanography, 57(3):881–894, 2012.
   doi: 10.4319/lo.2012.57.3.0881.

- R. J. Lynn and J. L. Reid. Characteristics and circulation of deep and abyssal waters. *Deep Sea Research and Oceanographic Abstracts*, 15(5):577–598, 1968. ISSN 0011-7471. doi: 10.1016/0011-7471(68)90064-8.
- T. J. McDougall. Neutral Surfaces. Journal of Physical Oceanography, 17(11):1950 1964,
   1987a. doi: 10.1175/1520-0485(1987)017<1950:NS>2.0.CO;2.
- T. J. McDougall. Thermobaricity, cabbeling, and water-mass conversion. *Journal of Geo*physical Research: Oceans, 92(C5):5448–5464, 1987b. doi: 10.1029/JC092iC05p05448.
- T. J. McDougall. Review of the Ocean Science manuscript egusphere-2025-3359 "Quantifying energy barriers associated with density stratification in vertical displacements of water parcels". EGUsphere, 2025. doi: 10.5194/egusphere-2025-3359-RC1.
- T. J. McDougall and P. M. Barker. Getting started with TEOS-10 and the Gibbs Seawater (GSW) oceanographic toolbox. Scor/iapso~WG,~127(532):1-28,~2011.
- E. Moreles, E. Romero, K. Ramos-Musalem, and L. Tenorio-Fernandez. The global ocean mixed layer depth derived from an energy approach based on buoyancy work. *Ocean Science*, 21(5):2019–2039, 2025. doi: 10.5194/os-21-2019-2025.
- M. M. Omand, R. Govindarajan, J. He, and A. Mahadevan. Sinking flux of particulate organic matter in the oceans: Sensitivity to particle characteristics. *Scientific Reports*, 10 (1):5582, 2020. doi: 10.1038/s41598-020-60424-5.
- B. G. Reichl, A. Adcroft, S. M. Griffies, and R. Hallberg. A Potential Energy Analysis of Ocean Surface Mixed Layers. *Journal of Geophysical Research: Oceans*, 127(7): e2021JC018140, 2022. doi: 10.1029/2021JC018140.
- J. L. Reid and R. J. Lynn. On the influence of the Norwegian-Greenland and Weddell seas upon the bottom waters of the Indian and Pacific oceans. *Deep Sea Research and Oceanographic Abstracts*, 18(11):1063–1088, 1971. ISSN 0011-7471. doi: 10.1016/0011-7471(71)90094-5.
- B. Rosenthal and F. Roquet. The Center of Mass of the Ocean as an Index of the General Stratification and Its Relation to the Overturning Circulation. *Journal of Physical Oceanography*, 55(3):277 291, 2025. doi: 10.1175/JPO-D-24-0078.1.
- W. Schmidt. Über die Temperatur- und Stabilitätsverhältnisse von Seen. Geografiska Annaler, 10:145-177, 1928. ISSN 16513215. doi: 10.2307/519789.
- J. H. Simpson and D. Bowers. Models of stratification and frontal movement in shelf seas.
   Deep Sea Research Part A. Oceanographic Research Papers, 28(7):727–738, 1981. ISSN 0198-0149. doi: 10.1016/0198-0149(81)90132-1.

- J. H. Simpson, C. M. Allen, and N. C. G. Morris. Fronts on the continental shelf. *Journal of Geophysical Research: Oceans*, 83(C9):4607–4614, 1978. doi: 10.1029/JC083iC09p04607.
- J. Sprintall and M. Tomczak. Evidence of the barrier layer in the surface layer of the tropics. *Journal of Geophysical Research: Oceans*, 97(C5):7305–7316, 1992. doi: 10.1029/92JC00407.
- G. Sutherland, G. Reverdin, L. Marié, and B. Ward. Mixed and mixing layer depths in the ocean surface boundary layer under conditions of diurnal stratification. *Geophysical Research Letters*, 41(23):8469–8476, 2014. doi: 10.1002/2014GL061939.
- G. K. Vallis. Atmospheric and Oceanic Fluid Dynamics: Fundamentals and Large-Scale Circulation. Cambridge University Press, 1 edition, 2006.
- E. M. Vincent, M. Lengaigne, J. Vialard, G. Madec, N. C. Jourdain, and S. Masson. Assessing the oceanic control on the amplitude of sea surface cooling induced by tropical cyclones.

  Journal of Geophysical Research: Oceans, 117(C5), 2012. doi: 10.1029/2011JC007705.
- X. Wang, G. Han, Y. Qi, and W. Li. Impact of barrier layer on typhoon-induced sea surface cooling. *Dynamics of Atmospheres and Oceans*, 52(3):367–385, 2011. ISSN 0377-0265. doi: 10.1016/j.dynatmoce.2011.05.002.
- Y. Yan, L. Li, and C. Wang. The effects of oceanic barrier layer on the upper ocean response to tropical cyclones. *Journal of Geophysical Research: Oceans*, 122(6):4829–4844, 2017. doi: 10.1002/2017JC012694.
- A. Zampollo, T. Cornulier, R. O'Hara Murray, J. F. Tweddle, J. Dunning, and B. E. Scott. The bottom mixed layer depth as an indicator of subsurface Chlorophyll *a* distribution. *Biogeosciences*, 20(16):3593–3611, 2023. doi: 10.5194/bg-20-3593-2023. URL https://bg.copernicus.org/articles/20/3593/2023/.

---

## Author Comment (AC2)

**Discussion of the manuscript: <a href="https://doi.org/10.5194/egusphere-2025-3359">https://doi.org/10.5194/egusphere-2025-3359</a>**

**Title: Quantifying energy barriers associated with density stratification in vertical displacements of water parcels**

Dear Dr. Griffies. We appreciate the time you take to review our preprint, the detailed mathematical derivation of the buoyant force, and the pertinent comments regarding our approach. As you suggested, the analysis regarding the net force on a displaced fluid parcel was carefully revised and corrected; we provide a better contextualization of our formulation and its limitations. Also, we revised the Introduction, incorporated recent relevant literature, and clarified the objective of our study. We hope you find the corrections satisfactory and that our research contributes to the advancement of oceanographic knowledge.

**Reviewer #2 Stephen M. Griffies**

**Comment 1:** Inaccurate theoretical foundation. In particular, the use of potential density is problematic when considering bulk measures. Vallis noted this limitation two pages later than the page 92 quoted by Moreles as their starting point. McDougall's review provides more points to this regard.

I concur with McDougall's concern for the use of potential density as the foundation for a bulk method to measure vertical stratification in the ocean. Granted, in many areas of the ocean there are no worries. But the high latitudes are a place where problems can occur. Instead of starting from first principles, the authors pick an equation from Vallis (2006) that is not even the final expression he ends up with to measure stratification. Vallis (2006) concludes on his page 94 that a local measure of stratification is the vertical derivative of the locally referenced potential density. I appreciate that Moreles et al are seeking a bulk measure rather than a local measure. But when one goes bulk, more theoretical work is needed to formulate the theory and then to test its relevance in the real ocean..

**Answer 1:** We agree with you that the use of potential density referenced to a fixed pressure is not adequate to measure integrated stratification across large distances since its use ignores the thermobaric effect. We apologize for the lack of clarity in stating the objective of our study; we don't intend to propose an index of stratification or a proxy for it. Our objective is to estimate the energy barriers associated with density stratification in vertical displacements of water parcels, considering the direction (upward or downward) in which the parcel is displaced. We suggest that the magnitude of the energy barriers can serve as a proxy for the intensity of stratification experienced by a water parcel as it displaces vertically; the greater the barriers, the more stratified the water column, and vice versa.

We agree with you on the relevance of describing the theoretical context associated with a physical problem. Therefore, in the revised version of the methodology (see the attached document), we described the problem and the approach we took to solve it:

To analyze the energy barriers, we quantified the buoyancy energy required for a water parcel in equilibrium to be displaced upward and downward. Given a vertical density stratification, we first derive an expression to estimate the force F on a water parcel, initially at rest, if it were displaced vertically (upward or downward) from its equilibrium position. If the force is conservative, we can

use classical mechanics arguments to obtain a potential function V from which the force F can be derived (Goldstein, 1980, p. 4),  $F = -\operatorname{grad} V$ . The potential function V is the buoyancy energy and is a function of the vertical coordinate. It represents the energy barriers a water parcel would encounter along its vertical movement. In summary, we need an expression of the force F from which we can obtain its associated potential V.

We began with the description of the physical setting of the problem; then, we presented the accurate expression, derived from first principles, to calculate the net force given. The detailed description of this physical setting and the derivation of the net force can be found in sections 2.9.1 and 2.9.2 of Vallis (2006) and McDougall (2025); this is the reason we did not provide the derivation of this expression. The accurate expression of the force was not the final one we used to calculate the potential function. We revised our calculations and described the different approaches we took to obtain the final expression of the force, which is an approximated expression. Using real ocean profiles, we examined the accuracy of the approximated expression for the force and found that calculating the force with the potential density of the environment referenced to a fixed pressure, centered at the section of interest, is sufficient for qualitative oceanography. The integrated error (measured in terms of the mean absolute percentage error) is less than 5% for pressure variations not exceeding 2000 dbar.

From the corrected expression for the force, we were able to define the buoyancy potential energy (BPE) to estimate the energy barriers associated with density stratification in vertical displacements of water parcels. We also calculated the energy required to vertically displace an insulated water parcel using the expression for BPE and the algorithms in the TEOS-10 toolbox suggested by McDougall (2025), considering real ocean profiles. As with the force, the integrated error at each depth (measured in terms of the mean absolute percentage error) is less than 5% in vertical sections with pressure variations of up to 2000 dbar.

**Comment 2:** Unclear use of physics. Writing down a math object that has physical dimensions does not make it physical. This point is relevant to the claim by Moreles et al that they are writing down forces and energies when, however, they never write the corresponding equation of motion and equation of energy.

As a further critique, I must admit to being disappointed when authors write down a mathematical expression that has physical dimensions and then claim those expressions are physically relevant without showing their corresponding equations of motion. There have been numerous attempts at providing a rational theory for ocean energetics, and they all get quite involved conceptually and mathematically. All of these details are relevant to the present question. One cannot presume to be presenting a new diagnostic without placing it within a theoretical context. Quite simply, stating that one is discussing forces and energies, without showing how they appear from first principles arguments, is not physics.

**Answer 2:** We revised the description of the physical problem and its associated calculations to clarify the details. *Please see the answer to Comment 1*.

**Comment 3:** Incomplete literature survey. The recent literature has been discussing energetic foundations for diagnosing vertical stratification. Moreles et al need to clearly identify the novelty of their manuscript. The papers by Reichl et al (2022) and Rosenthal and Roquet (2025) are central to the

proposed Moreles et al approach. Reichl et al is cited by Moreles, but no theoretical or practical comparison is provided. Rosenthal and Roquet is not cited.

As coauthor of the Reichl et al (2022) potential energy paper, I was disappointed that Moreles et al did not provide any specific comment on the novelty of their proposed method relative to Reichl et al. Indeed, Reichl et al develop a potential energy approach. They acknowledge the difficulties with the seawater equation of state and then test approximations that make use of potential density. The Reichl approach satisfies the goals of Moreles et al. by providing a bulk enegetics approach that only requires hydrographic data. So where does Reichl et al fail where Moreles et al succeed?

I was also surprised by the absence of the Rosenthal and Roquet (2025) paper, DOI: 10.1175/JPO-D-24-0078.1, who propose a center of mass approach to defining stratification. This approach has some relation to the Reichl et al potential energy approach.

Answer 3: We clarified the objective of our study; now, the literature review is relevant to the gap we try to fill. The gap we identified is in the estimation of energy barriers associated with density stratification in the vertical displacements of water parcels, considering the direction (upward or downward) in which the parcel is displaced. The studies by Reichl et al. (2022) and Rosenthal and Roquet (2025) provide an interesting description of ocean energetics; their approaches may shape the new standard for analyzing energy and stratification in the ocean. Our proposal does not intend to replace or repair any failures in the studies of Reichl et al. (2022) and Rosenthal and Roquet (2025), as our objectives differ. Instead, our study aims to complement traditional analyses of the ocean's dynamics and thermodynamics by providing a complementary variable to characterize the vertical structure of the water column.

We thank the reviewer for letting us know about Rosenthal and Roquet's paper. This paper was published on March 21, 2025, which roughly coincided with the date we submitted our manuscript for possible publication in a scientific journal. It is by no means our intention to deliberately or carelessly omit relevant literature.

**Comment 4:** Minor point: I found the use of acronyms to be excessive. In particular, there are many acronyms in the conclusion that are not defined within the conclusion. Many readers read the abstract than conclusions in that order, skipping the intermediate sections. Having undefined acronyms in the conclusion makes it tough to understand the main points of the manuscript. I strongly recommend greatly reducing the use of acronyms, except where they are community standards.

**Answer 4:** We apologize for the writing style in the Conclusions section. It will be corrected in the revised version of the manuscript if we are allowed to submit it.

**1 1 Introduction**

The vertical variation in density in aquatic bodies (i.e., oceans, seas, freshwater bodies, and estuaries), resulting from temperature and salinity differences, constitutes their stratification, which has significant implications for the variability of weather and climate processes as it determines different dynamical, ecological, chemical, and biological processes occurring there. Stratification has a significant role on the flux of particulate organic matter from the surface to sediments (e.g., Kirillin et al., 2012; Omand et al., 2020), the vertical content of chlorophyll (e.g., Cullen, 2015; Carvalho et al., 2017; Briseño-Avena et al., 2020; Cornec et al., 2021; Zampollo et al., 2023), biological productivity (e.g., Franks, 2014; Bouman et al., 2020), the mixed layer (e.g., Sutherland et al., 2014; Gray et al., 2020; Moreles et al., 2025), the 10 ocean-atmosphere coupling and exchanges between them (e.g., Deser et al., 2010; Groeskamp 11 et al., 2019), barrier layers (e.g., Sprintall and Tomczak, 1992; Cronin and McPhaden, 2002), 12 and tropical cyclone dynamics (e.g., Wang et al., 2011; Balaguru et al., 2012; Vincent et al., 13 2012; Yan et al., 2017). 14

15 16

17

18

19

20

21

23

24

25

27

29

31

32

33

34

35

36

37

The vertical structure of aquatic bodies in terms of their stratification and energetics has been intensively studied. Schmidt (1928) derived an equation for the stability of a lake; Idso (1973) subsequently extended that equation to account for the contribution of each lake layer to stability. Simpson et al. (1978) and Herrmann et al. (2008) proposed energy-based indexes to quantify stratification from the surface to a specific depth. Dynamic equations derived from Simpson et al.'s index are used to analyze the evolution of stratification in terms of different processes (Simpson and Bowers, 1981; Burchard and Hofmeister, 2008; de Boer et al., 2008). Using boundary-layer turbulence theory, Reichl et al. (2022) proposed the potential energy anomaly of the water column to estimate the depth to which a given energy could homogenize a layer of seawater; from this, they introduced a framework for diagnosing the ocean mixed layer depth. The potential energy anomaly provides a proxy for the stratification of a layer of seawater by estimating its energetic distance from a well-mixed state. From energetic foundations, Rosenthal and Roquet (2025) developed an index of stratification strength for the global ocean using the height anomaly, defined as the height of the ocean's center of mass relative to the height of a fully-mixed state. The lower the center of mass, the larger the stratification, and vice versa. They also developed a tendency equation for the budget of the height anomaly, which helps identify the overall contribution of different processes to the stratification. Moreles et al. (2025) proposed the buoyancy work required to displace a water parcel vertically as a proxy for the vertical homogeneity of the water column: the lower the work, the greater the vertical homogeneity of the water column, and vice versa. From this, they defined the ocean mixed layer and computed a global monthly climatology of its depth that maintains quasi-homogeneity in energy, density, and temperature along the mixed layer year-round.

38 39 40

Stratification determines mixing, which is commonly associated with the interchange of properties between deep and shallow waters. This vertical interchange of properties is typically linked to the vertical displacement of water parcels, which carry their properties from one isopycnal to another. The vertical movement of a water parcel is, to a certain extent, determined by the density stratification experienced by the water parcel when it displaces upward and downward; the more stratified the water column, the more energy is required to displace the water parcel vertically, and vice versa. What are the characteristics of the energy barriers associated with density stratification in vertical displacements of water parcels? The above is particularly relevant when studying processes (physical, ecological, chemical, or biological) that occur in the subsurface or at a distance from the surface, where the direction of vertical movement of water parcels is relevant. For any intermediate section in the water column, the energy barriers within it depend on the direction (upward or downward) in which they are measured: the barriers experienced by a deep-water parcel when it displaces upward differ from those experienced by a shallow-water parcel when it displaces downward.

The above question has not been addressed in previous studies; thus, our objective is to estimate the energy barriers associated with density stratification in vertical displacements of water parcels. For a water parcel at any depth within the water column, we estimate the energy barriers it would encounter if vertically displaced upward or downward. The magnitude of the energy barriers to vertical displacement of a water parcel can serve as a proxy for the intensity of stratification it experiences along its vertical displacement; the greater the barriers, the more stratified the water column, and vice versa. By focusing on the direction of a water parcel's vertical displacement, we can analyze stratification in terms of the direction of quantification, thus enhancing the analyses of stratification in aquatic bodies. The metric used to estimate energy barriers is presented in Section 2, followed by its application to dynamical and ecological contexts in Section 3, which demonstrates its potential for enriching ocean analyses.

**67 2 Estimating energy barriers**

To analyze the energy barriers, we quantified the buoyancy energy required for a water parcel in equilibrium to be displaced upward and downward. Given a vertical density stratification, we first derive an expression to estimate the force F on a water parcel, initially at rest, if it were displaced vertically (upward or downward) from its equilibrium position. If the force is conservative (it is path-independent), we can use classical mechanics arguments to obtain a potential function V from which the force F can be derived (Goldstein, 1980, p. 4),

$$\vec{F} = -\nabla V. \tag{1}$$

The potential function V is the buoyancy energy and is a function of the vertical coordinate. It represents the energy barriers a water parcel would encounter along its vertical movement.

The physical setting of the problem involves a fluid in hydrostatic balance in a constant gravitational field, where rotation effects, horizontal motion, and friction are neglected. This

physical setting is typically used when analyzing the oscillation of a fluid perturbed away from its resting state, and the Brunt-Väisälä frequency is derived. For our derivation, we will start from the expression for the net force; for a detailed description of this physical setting and the derivation of the net force, see sections 2.9.1 and 2.9.2 of Vallis (2006) and McDougall (2025). A water parcel at its equilibrium position  $z_{eq}$  that is slowly vertically displaced from that level to any depth z, without exchanging either mass or heat with the surroundings, experiences a net force per unit volume given by (Vallis, 2006, p. 92),

$$F(z) = g\left[\hat{\rho}(z) - \rho(z)\right],\tag{2}$$

where g is the acceleration due to gravity,  $\hat{\rho}$  is the in-situ density of the environment, and  $\rho$  is the in-situ density of the parcel.

88

92

94

96

98

99

100

101

102

103

104

105

106

107

108

Since the parcel is undergoing adiabatic conditions during the displacement, the net force (Eq. 2) can be expressed as a function of pressure P as (McDougall, 2025),

$$g\{\hat{\rho}[S(P), \Theta(P), P] - \hat{\rho}[S_{eq}, \Theta_{eq}, P]\}$$
 force per unit volume, (3a)

$$g\left\{\frac{\hat{v}\left[S_{\text{eq}},\Theta_{\text{eq}},P\right]}{\hat{v}\left[S(P),\Theta(P),P\right]}-1\right\} \quad \text{force per unit mass,}$$
 (3b)

where  $\hat{v}$  is the specific volume of the environment (the reciprocal of  $\hat{\rho}$ ) and the absolute salinity and conservative temperature of the environment have been regarded as a function of P as S(P) and  $\Theta(P)$ , respectively. The subscript eq refers to the properties of the parcel at the equilibrium position  $z_{\rm eq}$  or  $P_{\rm eq}$ .

Equations (3), derived from first principles, represent the accurate expressions to calculate the net force on the water parcel when it is vertically displaced from its equilibrium position  $z_{\text{eq}}$  (or  $P_{\text{eq}}$ ) to any depth z (or P) under adiabatic conditions. However, they are not easily structured to calculate the associated potential function using Eq. (1) because the force is a composite function of pressure in terms of absolute salinity and conservative temperature. To deal with this problem, we explored an approximation for the force, valid for small displacements, calculated in terms of the potential density of the environment referenced to the pressure at the level z (Vallis, 2006, p. 93),

$$F(z) \approx g \left[ \rho_{\theta}(z) - \rho_{\theta}(z_{\text{eq}}) \right],$$
 (4)

where  $\rho_{\theta}$  is the locally-referenced potential density. Eq. (4) is still insufficient to calculate the associated potential function due to the potential density is not referenced to a fixed pressure, which results in that the vertical coordinate is not unique throughout the displacement. In order to calculate the associated potential function using Eq. (1), we must calculate the force with Eq. (4) but using the potential density referenced to a fixed pressure. However, the use of potential density referenced to a fixed pressure ignores the thermobaric effect, which can lead to significant errors for large vertical displacements of the water parcel (McDougall,

1987a,b). Thus, it is necessary to examine whether this approximation of the force is accurate enough to replace the accurate expression (Eqs. 3), at least for qualitative oceanography.

We selected three Argo profiles (first column of Fig. 1), exhibiting different stratification conditions, to analyze the differences between various versions of the force: (i) the accurate expression given by Eq. (3b) and the approximate expressions given by Eq. (4), considering (ii) the locally-referenced potential density and (iii) the potential density referenced to a fixed pressure. For each profile, we calculated the forces at various depths, considering that the equilibrium position  $P_{\rm eq}$  of the water parcel is at the isothermal layer depth, defined as the depth at which the conservative temperature has decreased by 0.2°C from the temperature at a depth of 10 m (de Boyer Montégut et al., 2004). Then, we calculated the differences between the approximate forces and the accurate force. Finally, we calculated the mean absolute percentage error (MAPE) between each approximate force and the accurate one at various depths; from  $P_{\rm eq}$ , we selected a vertical section that increased in length (both upward and downward) to calculate the MAPE. We used the Thermodynamic Equation of SeaWater 2010 (McDougall and Barker, 2011) to calculate the different variables in the equations of interest.

A visual inspection of the results shows that the calculated force is very similar across all versions (second column of Fig. 1). At each depth, the differences between the approximate forces with respect to the exact force are three orders of magnitude smaller if the locally-referenced potential density is used and one order of magnitude smaller if the potential density is referenced to a fixed pressure (third column of Fig. 1). The differences at  $P_{\rm eq}$  are zero and they increase for depths far from  $P_{\rm eq}$  and far from the reference pressure used to calculate the potential density. However, since we are interested in integrated measures, the MAPE at various depths is a better measure to quantify the error of the approximate forces (fourth column of Fig. 1). The force calculated with Eq. (4) using the locally-referenced potential density is nearly the same as the accurate one, with MAPE of less than 1% throughout the vertical. The MAPE can be very large when using the potential density referenced to a fixed pressure far from  $P_{\rm eq}$ , even if the vertical displacements of the parcel are small; the MAPE can reach up to 30% when the differences between  $P_{\rm eq}$  and the reference pressure exceed 1500 dbar. When using Eq. (4) with the potential density referenced to  $P_{\rm eq}$ , the MAPE values are less than 5% throughout the vertical (with pressure variations of up to 2000 dbar).

Using inductive reasoning, we assume that the above results are maintained for the world ocean with pressure variations of 2000 dbar, suggesting the following. The approximate force (Eq. 4) calculated with the potential density referenced to a fixed pressure, centered in the section of interest, is sufficient for qualitative oceanography; the integrated error will presumably be less than 5% for pressure variations not exceeding 2000 dbar. This suggestion is in agreement with the findings of Lynn and Reid (1968) and Reid and Lynn (1971), who observed that if the vertical section of interest does not exceed pressure variations of about 1000 dbar, the stability of the water column is adequately described with the potential den-

Figure 1: **First column:** ocean profiles in the Pacific, Southern, and Atlantic Oceans; the profiles of potential density anomaly  $\sigma$  are referenced to  $P_{\rm eq}$ . **Second column:** accurate net force and approximations to it calculated with different versions of potential density. **Third column:** differences between the approximate forces and the accurate force. **Fourth column:** the MAPE of each approximate force at various depths, calculated from  $P_{\rm eq}$ .

sity referenced to a pressure centered in the section of interest. Despite the high accuracy expected when using Eq. (4), we can always calculate the accurate force via Eqs. (3) and quantify the integrated error associated with using the approximate expression.

The above justify using the approximate expression (Eq. 4) to calculate the net force on the water parcel when it is vertically displaced from its equilibrium position  $z_{eq}$  to any depth z. Since we are using the potential density referenced to a fixed pressure, the vertical coordinate is unique throughout the displacement and we can calculate the potential energy function associated with the net buoyant force, the buoyancy potential energy (BPE), using

$$F(z) = -\frac{\mathrm{d}}{\mathrm{d}z} \mathrm{BPE}(z). \tag{5}$$

BPE is obtained by vertically integrating the force in Eq. (4),

Eq. (1),

$$BPE(z) = BPE(z_{eq}) - \int_{z_{eq}}^{z} g \left[ \rho_{\theta}(\gamma) - \rho_{\theta}(z_{eq}) \right] d\gamma = BPE(z_{eq}) + g(z - z_{eq}) \rho_{\theta}(z_{eq}) - g \int_{z_{eq}}^{z} \rho_{\theta}(\gamma) d\gamma,$$
(6)

where  $\rho_{\theta}$  is the potential density of the environment referenced to a fixed pressure, centered in the section of interest. When working with potentials, the physically relevant quantity is the potential difference between two depths; thus, we can set BPE( $z_{eq}$ ) = 0 without loss of generality. BPE represents the energy barriers a water parcel would encounter if it were displaced from its equilibrium position  $z_{eq}$  to any depth z. The work done by the net buoyant force in displacing a water parcel from  $z_1$  to  $z_2$  is BPE( $z_1$ ) – BPE( $z_2$ ). BPE is directly related to the work done by buoyancy proposed by Moreles et al. (2025); thus, all the properties and attributes of the work done by buoyancy are directly applicable to BPE.

The expression given by Eq. (6) is an approximate expression for the total energy required to slowly move an insulated parcel of fluid from its equilibrium location to any final location. The accurate expression is given by vertically integrating the force in Eqs. (3), as shown by McDougall (2025) in his Eqs. (3) and (4). Through a meticulous and detailed review of this preprint, McDougall (2025) identified a connection between BPE and the Cunningham geostrophic streamfunction, a novel result. He then proposed a way to calculate this energy using the TEOS-10 Toolbox (see his Eq. 5). We computed the energy for each profile shown in Fig. 1 using BPE (Eq. 6) and Eq. 5 of McDougall (2025) (plots not shown). Similar to what we found in the force analysis, the differences in the energy values at each depth obtained with these two expressions are minimal (the MAPE between them is less than 5% throughout the vertical), suggesting that BE is accurate enough for calculating the energy in vertical sections not exceeding pressure variations of 2000 dbar. Again, we can always calculate the energy using the accurate expression and quantify the integrated error associated with using the approximate expression.

Our approach provides a physically derived, approximated, and intuitive variable (i.e.,

BPE) to estimate the energy barriers associated with density stratification in vertical displacements of water parcels, which is accurate enough for qualitative oceanography. Depending on the sign of BPE, two physical situations are identified. For BPE > 0, the force and the parcel displacement are in opposite directions, causing the parcel to decelerate when ... ... it continues as it is in the preprint from line 68.

192

Note: According to the new way for calculating BPE, Fig. 3 regarding the barrier layers will be adjusted to reflect the BPE calculated using the potential density referenced to a fixed pressure, centered between the mixed layer depth and the isothermal layer depth. The prior BPE and the new BPE are nearly identical; therefore, the discussion and results of the new figure are maintained as in the preprint version with this adjustment.

**References**

- K. Balaguru, P. Chang, R. Saravanan, L. R. Leung, Z. Xu, M. Li, and J.-S. Hsieh. Ocean barrier layers' effect on tropical cyclone intensification. *Proceedings of the National Academy of Sciences*, 109(36):14343–14347, 2012. doi: 10.1073/pnas.1201364109.
- H. A. Bouman, T. Jackson, S. Sathyendranath, and T. Platt. Vertical structure in chlorophyll profiles: influence on primary production in the Arctic Ocean. *Philosophical Transac*tions of the Royal Society A: Mathematical, Physical and Engineering Sciences, 378(2181): 20190351, 2020. doi: 10.1098/rsta.2019.0351.
- C. Briseño-Avena, J. C. Prairie, P. J. S. Franks, and J. S. Jaffe. Comparing Vertical Distributions of Chl-a Fluorescence, Marine Snow, and Taxon-Specific Zooplankton in Relation to Density Using High-Resolution Optical Measurements. Frontiers in Marine Science, 7, 2020. ISSN 2296-7745. doi: 10.3389/fmars.2020.00602.
- H. Burchard and R. Hofmeister. A dynamic equation for the potential energy anomaly for analysing mixing and stratification in estuaries and coastal seas. *Estuarine*, *Coastal and Shelf Science*, 77(4):679–687, 2008. ISSN 0272-7714. doi: 10.1016/j.ecss.2007.10.025.
- F. Carvalho, J. Kohut, M. J. Oliver, and O. Schofield. Defining the ecologically relevant mixed-layer depth for Antarctica's coastal seas. *Geophysical Research Letters*, 44(1):338–345, 2017. doi: 10.1002/2016GL071205.
- M. Cornec, H. Claustre, A. Mignot, L. Guidi, L. Lacour, A. Poteau, F. D'Ortenzio, B. Gentili,
   and C. Schmechtig. Deep Chlorophyll Maxima in the Global Ocean: Occurrences, Drivers
   and Characteristics. Global Biogeochemical Cycles, 35(4):e2020GB006759, 2021. doi: 10.
   1029/2020GB006759. e2020GB006759 2020GB006759.
- M. F. Cronin and M. J. McPhaden. Barrier layer formation during westerly wind bursts.

  Journal of Geophysical Research: Oceans, 107(C12):SRF 21–1–SRF 21–12, 2002. doi: 10.1029/2001JC001171.

- J. J. Cullen. Subsurface Chlorophyll Maximum Layers: Enduring Enigma or Mystery Solved?
   Annual Review of Marine Science, 7(Volume 7, 2015):207–239, 2015. ISSN 1941-0611. doi:
   10.1146/annurev-marine-010213-135111.
- G. J. de Boer, J. D. Pietrzak, and J. C. Winterwerp. Using the potential energy anomaly equation to investigate tidal straining and advection of stratification in a region of freshwater influence. *Ocean Modelling*, 22(1):1–11, 2008. ISSN 1463-5003. doi: 10.1016/j.ocemod.2007.12.003.
- C. de Boyer Montégut, G. Madec, A. S. Fischer, A. Lazar, and D. Iudicone. Mixed layer depth over the global ocean: An examination of profile data and a profile-based climatology.
   Journal of Geophysical Research: Oceans, 109(C12), 2004. doi: 10.1029/2004JC002378.
- C. Deser, M. A. Alexander, S.-P. Xie, and A. S. Phillips. Sea Surface Temperature Variability:
   Patterns and Mechanisms. Annual Review of Marine Science, 2(Volume 2, 2010):115–143,
   2010. ISSN 1941-0611. doi: 10.1146/annurev-marine-120408-151453.
- P. J. S. Franks. Has Sverdrup's critical depth hypothesis been tested? Mixed layers vs. turbulent layers. *ICES Journal of Marine Science*, 72(6):1897–1907, 10 2014. ISSN 1054-3139. doi: 10.1093/icesjms/fsu175.
- H. Goldstein. *Classical Mechanics*. Addison-Wesley series in physics. Addison-Wesley, Reading, Mass., 2nd ed edition, 1980.
- E. Gray, E. B. Mackay, J. A. Elliott, A. M. Folkard, and I. D. Jones. Wide-spread inconsistency in estimation of lake mixed depth impacts interpretation of limnological processes.

  Water Research, 168:115136, 2020. ISSN 0043-1354. doi: 10.1016/j.watres.2019.115136.
- S. Groeskamp, S. M. Griffies, D. Iudicone, R. Marsh, A. G. Nurser, and J. D. Zika. The Water Mass Transformation Framework for Ocean Physics and Biogeochemistry. *Annual Review of Marine Science*, 11(Volume 11, 2019):271–305, 2019. ISSN 1941-0611. doi: 10.1146/annurev-marine-010318-095421.
- M. Herrmann, S. Somot, F. Sevault, C. Estournel, and M. Déqué. Modeling the deep convection in the northwestern Mediterranean Sea using an eddy-permitting and an eddy-resolving model: Case study of winter 1986–1987. *Journal of Geophysical Research:*Oceans, 113(C4), 2008. doi: 10.1029/2006JC003991.
- S. B. Idso. On the concept of lake stability. *Limnology and Oceanography*, 18(4):681–683, 1973. doi: 10.4319/lo.1973.18.4.0681.
- G. Kirillin, H.-P. Grossart, and K. W. Tang. Modeling sinking rate of zooplankton carcasses:
   Effects of stratification and mixing. Limnology and Oceanography, 57(3):881–894, 2012.
   doi: 10.4319/lo.2012.57.3.0881.

- R. J. Lynn and J. L. Reid. Characteristics and circulation of deep and abyssal waters. *Deep Sea Research and Oceanographic Abstracts*, 15(5):577–598, 1968. ISSN 0011-7471. doi: 10.1016/0011-7471(68)90064-8.
- T. J. McDougall. Neutral Surfaces. Journal of Physical Oceanography, 17(11):1950 1964,
   1987a. doi: 10.1175/1520-0485(1987)017<1950:NS>2.0.CO;2.
- T. J. McDougall. Thermobaricity, cabbeling, and water-mass conversion. *Journal of Geo*physical Research: Oceans, 92(C5):5448–5464, 1987b. doi: 10.1029/JC092iC05p05448.
- T. J. McDougall. Review of the Ocean Science manuscript egusphere-2025-3359 "Quantifying energy barriers associated with density stratification in vertical displacements of water parcels". EGUsphere, 2025. doi: 10.5194/egusphere-2025-3359-RC1.
- T. J. McDougall and P. M. Barker. Getting started with TEOS-10 and the Gibbs Seawater (GSW) oceanographic toolbox. Scor/iapso~WG,~127(532):1-28,~2011.
- E. Moreles, E. Romero, K. Ramos-Musalem, and L. Tenorio-Fernandez. The global ocean mixed layer depth derived from an energy approach based on buoyancy work. *Ocean Science*, 21(5):2019–2039, 2025. doi: 10.5194/os-21-2019-2025.
- M. M. Omand, R. Govindarajan, J. He, and A. Mahadevan. Sinking flux of particulate organic matter in the oceans: Sensitivity to particle characteristics. *Scientific Reports*, 10 (1):5582, 2020. doi: 10.1038/s41598-020-60424-5.
- B. G. Reichl, A. Adcroft, S. M. Griffies, and R. Hallberg. A Potential Energy Analysis of Ocean Surface Mixed Layers. *Journal of Geophysical Research: Oceans*, 127(7): e2021JC018140, 2022. doi: 10.1029/2021JC018140.
- J. L. Reid and R. J. Lynn. On the influence of the Norwegian-Greenland and Weddell seas upon the bottom waters of the Indian and Pacific oceans. *Deep Sea Research and Oceanographic Abstracts*, 18(11):1063–1088, 1971. ISSN 0011-7471. doi: 10.1016/0011-7471(71)90094-5.
- B. Rosenthal and F. Roquet. The Center of Mass of the Ocean as an Index of the General Stratification and Its Relation to the Overturning Circulation. *Journal of Physical Oceanography*, 55(3):277 291, 2025. doi: 10.1175/JPO-D-24-0078.1.
- W. Schmidt. Über die Temperatur- und Stabilitätsverhältnisse von Seen. Geografiska Annaler, 10:145-177, 1928. ISSN 16513215. doi: 10.2307/519789.
- J. H. Simpson and D. Bowers. Models of stratification and frontal movement in shelf seas.
   Deep Sea Research Part A. Oceanographic Research Papers, 28(7):727–738, 1981. ISSN 0198-0149. doi: 10.1016/0198-0149(81)90132-1.

- J. H. Simpson, C. M. Allen, and N. C. G. Morris. Fronts on the continental shelf. *Journal of Geophysical Research: Oceans*, 83(C9):4607–4614, 1978. doi: 10.1029/JC083iC09p04607.
- J. Sprintall and M. Tomczak. Evidence of the barrier layer in the surface layer of the tropics. *Journal of Geophysical Research: Oceans*, 97(C5):7305–7316, 1992. doi: 10.1029/92JC00407.
- G. Sutherland, G. Reverdin, L. Marié, and B. Ward. Mixed and mixing layer depths in the ocean surface boundary layer under conditions of diurnal stratification. *Geophysical Research Letters*, 41(23):8469–8476, 2014. doi: 10.1002/2014GL061939.
- G. K. Vallis. Atmospheric and Oceanic Fluid Dynamics: Fundamentals and Large-Scale Circulation. Cambridge University Press, 1 edition, 2006.
- E. M. Vincent, M. Lengaigne, J. Vialard, G. Madec, N. C. Jourdain, and S. Masson. Assessing the oceanic control on the amplitude of sea surface cooling induced by tropical cyclones.

  Journal of Geophysical Research: Oceans, 117(C5), 2012. doi: 10.1029/2011JC007705.
- X. Wang, G. Han, Y. Qi, and W. Li. Impact of barrier layer on typhoon-induced sea surface cooling. *Dynamics of Atmospheres and Oceans*, 52(3):367–385, 2011. ISSN 0377-0265. doi: 10.1016/j.dynatmoce.2011.05.002.
- Y. Yan, L. Li, and C. Wang. The effects of oceanic barrier layer on the upper ocean response to tropical cyclones. *Journal of Geophysical Research: Oceans*, 122(6):4829–4844, 2017. doi: 10.1002/2017JC012694.
- A. Zampollo, T. Cornulier, R. O'Hara Murray, J. F. Tweddle, J. Dunning, and B. E. Scott. The bottom mixed layer depth as an indicator of subsurface Chlorophyll *a* distribution. *Biogeosciences*, 20(16):3593–3611, 2023. doi: 10.5194/bg-20-3593-2023. URL https://bg.copernicus.org/articles/20/3593/2023/.